# On the Robustness of Langevin Dynamics to Score Function Error

**Daniel Yiming Cao** [* 1]  **August Y. Chen** [* 1]  **Karthik Sridharan** [* 1]  **Yuchen Wu** [* 2]

## Abstract

We consider the robustness of score-based generative modeling to errors in the estimate of the score function. In particular, we show that Langevin dynamics is not robust to the $L^2$ errors (more generally $L^p$ errors) in the estimate of the score function. It is well-established that with small $L^2$ errors in the estimate of the score function, diffusion models can sample faithfully from the target distribution under fairly mild regularity assumptions in a polynomial time horizon. In contrast, our work shows that even for simple distributions in high dimensions, Langevin dynamics run for any polynomial time horizon will produce a distribution far from the target distribution in Total Variation (TV) distance, even when the $L^2$ error (more generally $L^p$) of the estimate of the score function is arbitrarily small. Considering such an error in the estimate of the score function is unavoidable in practice when learning the score function from data, our results provide further justification for diffusion models over Langevin dynamics and serve to caution against the use of Langevin dynamics with estimated scores.

## 1. Introduction

Many sampling algorithms ubiquitous in statistics and Machine Learning (ML) – used from Bayesian inference to modern generative modeling – are score-based sampling algorithms. These sampling algorithms are widely used in a range of scientific and engineering applications, such as image generation (see e.g. Croitoru et al. (2023)), inverse problems in applied mathematics (e.g. Sanz-Alonso et al. (2023)), physical sciences (e.g. Zheng et al. (2025)), protein design and computational biology (e.g. Guo et al. (2024)),

and medical image reconstruction (e.g. Chung et al. (2022)). Many such score-based or related sampling algorithms exist. An incomplete list includes Langevin dynamics (see e.g. Welling & Teh (2011); Durmus et al. (2018); Chewi (2025)), diffusion models (e.g. Ho et al. (2020); Song & Ermon (2019); Song et al. (2021a); Song & Ermon (2020); Song et al. (2021b) for some foundational early works), flow matching (e.g. Lipman et al. (2023)), and stochastic interpolants (e.g. Albergo et al. (2025)).

Among these sampling algorithms, we investigate Langevin dynamics and diffusion models. Langevin dynamics is a classical technique for sampling from target distributions in statistics and ML, while diffusion models are a modern, popular and effective method for generative AI. At a high level, these algorithms approximately sample from a target distribution $\pi_{\mathrm{tar}}$ in $\mathbb{R}^d$ by running a stochastic process driven by the *score function* $\nabla \log \pi_{\mathrm{tar}}$ in the case of Langevin dynamics, or a suitable sequence of score functions $\nabla \log \pi_0, \nabla \log \pi_1, \ldots, \nabla \log \pi_k$ (the *annealed score functions*) in the case of diffusion models, where $\pi_0 \approx \pi_{\mathrm{tar}}$ and $\pi_0, \ldots, \pi_k$ forms a gradually noised version of the target distribution $\pi_{\mathrm{tar}}$. The continuous-time idealization of both these processes converges to $\pi_{\mathrm{tar}}$ under mild assumptions on $\pi_{\mathrm{tar}}$, see Section 2.2.

**Our focus:** In generative modeling applications, the score functions $\nabla \log \pi_{\mathrm{tar}}$ or $\nabla \log \pi_0$, $\nabla \log \pi_1$, ..., $\nabla \log \pi_k$ are not explicitly known and must be estimated from data (from $\pi_{\mathrm{tar}}$). This is done in practice by *score matching*: training a suitable parametric model (e.g., neural network) to learn these functions from data, by minimizing a suitable $L^2$ or similar loss. See, e.g., Ho et al. (2020); Song & Ermon (2019); Song et al. (2021a); Song & Ermon (2020); Song et al. (2021b); Karras et al. (2022); Chen et al. (2023b) for diffusion models, and Koehler et al. (2023); Pabbaraju et al. (2023); Koehler & Vuong (2024); Koehler et al. (2025) for Langevin dynamics. This yields estimates of the score function(s) that are accurate in, e.g., $L^2$. It is a priori unclear whether running Langevin dynamics or diffusion models with these estimates – as done in practice – leads to faithful samples from the target $\pi_{\mathrm{tar}}$ as happens with the respective idealized continuous-time processes. This motivates the fundamental question:

**Main Question:** *Is small $L^2$ (or more generally $L^p$) score*

Authors listed in alphabetical order. [1]Department of Computer Science, Cornell University, Ithaca, USA [2]School of Operations Research and Information Engineering, Cornell University, Ithaca, USA. Correspondence to: Daniel Yiming Cao <dyc33@cornell.edu>, August Y. Chen <ayc74@cornell.edu>.

*Proceedings of the 43rd International Conference on Machine Learning*, Seoul, South Korea. PMLR 306, 2026. Copyright 2026 by the author(s).

*estimation error with respect to $\pi_{\text{tar}}$ sufficient for the success of score-based sampling algorithms?*

This question has been answered affirmatively in the context of diffusion models (see e.g. Chen et al. (2023a); Benton et al. (2024); Li et al. (2024); Liang et al. (2025) among many more works) for general target distributions $\pi_{\text{tar}}$ with bounded second moment. Here we emphasize that this result for diffusion models assumes that a suitable weighted average of the $L^2$ score estimation errors of *all of* $\nabla \log \pi_0, \nabla \log \pi_1, \ldots, \nabla \log \pi_k$ (with respect to $\pi_0, \pi_1, \ldots, \pi_k$ respectively) is small, rather than just small $L^2$ estimation error of $\nabla \log \pi_{\text{tar}}$. In particular, let $\widehat{\pi_{\text{tar}}}$ denote the output distribution. With an estimated score $s$ and appropriate discretization, it is known that within a number of iterations **polynomial** in $d$, the sampling error scales linearly with the $L^2$ score estimation error:

$$\mathsf{TV}(\pi_{\text{tar}}, \widehat{\pi_{\text{tar}}}) \lesssim \varepsilon_{\text{score}},$$

where $\mathsf{TV}$ is Total Variation (TV) distance and $\varepsilon_{\text{score}}^2$ is a suitable weighted average of the $L^2$ score estimation errors of $\nabla \log \pi_0, \nabla \log \pi_1, \ldots, \nabla \log \pi_k$ with respect to $\pi_0, \pi_1, \ldots, \pi_k$ respectively. Such a result justifies the empirical success of diffusion models run with estimated score functions, such as those learned via score matching.

In sharp contrast, *the Main Question has not been answered for Langevin dynamics*. For an overview of related work:

- Several works such as Das et al. (2023); Huang et al. (2024) study the robustness of Langevin dynamics to $L^\infty$ bounds on the score estimate $\hat{s}$. However, $L^2$ and more generally $L^p$ bounds on score estimation error w.r.t. $\pi_{\text{tar}}$ are more realistic than $L^\infty$ bounds when learning $\hat{s}$ from data with score matching (Chen et al., 2023a; Koehler & Vuong, 2024; Koehler et al., 2025).[1] We note Lee et al. (2022) studies the robustness of Langevin dynamics with $L^2$ error bounds on $\hat{s}$. See their Theorem 2.1, which requires an error bound that is often *exponentially small in dimension*. Our Theorem 3.1 confirms the intuition in Lee et al. (2022), showing that their result is sharp up to the constant in the exponent.

- Langevin dynamics is a natural and widely used example of a Markov chain. Several works have studied the algorithmic robustness of Markov chains, dating as far back as Schweitzer (1968) to more recent investigations in Gaitonde & Mossel (2025); Zuckerman (2026). Our work performs a similar investigation in the context of score estimation error for the canonical Langevin dynamics Markov chain.

- Several works (Block et al., 2022; Lee et al., 2022; Xun et al., 2025) investigate the Main Question assuming $L^p$-accurate estimates to *all the annealed* score functions.[2] Note with $L^p$ accurate estimates to the *annealed* score functions, diffusion models are guaranteed to be successful by results of e.g. Chen et al. (2023b); Benton et al. (2024). Instead, our work focuses on when one only has an $L^2$ or $L^p$ accurate estimate of $\nabla \log \pi_{\text{tar}}$. We remark there is evidence that learning an accurate estimate of $\nabla \log \pi_{\text{tar}}$ can be computationally easier than learning all the annealed score functions for particular distributions $\pi_{\text{tar}}$ (Montanari & Vu, 2025; Koehler & Vuong, 2024).

- Several other works have studied statistical and computational aspects of score matching to estimate $\nabla \log \pi_{\text{tar}}$, such as Koehler et al. (2023); Pabbaraju et al. (2023); Montanari & Vu (2025). However these works do not answer the Main Question.

## 1.1. Main Results

In this work, **we strongly answer the Main Question in the negative for Langevin dynamics.** We show that an $L^2$-accurate (and more generally $L^p$-accurate) score estimate $\hat{s}$ to $\nabla \log \pi_{\text{tar}}$ with respect to $\pi_{\text{tar}}$ does not suffice for Langevin dynamics to successfully sample – this assumption on $L^2$ score estimation is most natural in the context of score estimates $\hat{s}$ that are learned from data (Chen et al., 2023a; Koehler & Vuong, 2024; Koehler et al., 2025). We establish:

1. **Theorem 3.1, Subsection 3.1:** Consider perhaps the most innocuous possible example, when $\pi_{\text{tar}}$ is an isotropic Gaussian in $\mathbb{R}^d$ and when we initialize Langevin dynamics at a standard Gaussian $N(0, I_d)$. When the estimated score $\hat{s}$ satisfies an arbitrarily small global $L^p$ bound $\mathbb{E}_{\pi_{\text{tar}}}[\|\hat{s} - \nabla \log \pi_{\text{tar}}\|^p]^{1/p}$ for any $p \geq 1$ in high dimensions, we construct examples where Langevin dynamics run with $\hat{s}$ from this natural initialization remains far from $\pi_{\text{tar}}$ for any poly$(d)$ time horizon in high dimensions. In particular, the TV distance between $\pi_{\text{tar}}$ and the law of Langevin dynamics run with $\hat{s}$ initialization from $N(0, I_d)$ is $1 - e^{-\Omega(d)}$. As a corollary, in high dimensions, the mixing time of Langevin dynamics run with $\hat{s}$ is also larger than any polynomial.[3]

2. **Theorem 3.7, Subsection 3.2:** We prove the phenomenon of Theorem 3.1 still applies when Langevin dynamics run with the score estimate $\hat{s}$ is initialized at a natural choice of warm start. Consider again

---

[1] In this paper, $L^2$ and more generally $L^p$ norms are implicitly taken with respect to $\pi_{\text{tar}}$ unless stated otherwise.

[2] Specifically when the annealed score functions are the convolution of the target distribution with a Gaussian.

[3] See Levin & Peres (2017) for a background on mixing time.

when the target distribution $\pi_{\mathrm{tar}}$ is an isotropic Gaussian in $\mathbb{R}^d$. Consider $n = \mathrm{poly}(d)$ i.i.d. samples from $\pi_{\mathrm{tar}}$. We construct a score estimate $\hat{s}$ *defined in terms of the $n$ i.i.d. samples* such that in high dimensions, $\hat{s}$ satisfies an arbitrarily small global $L^p$ bound $\mathbb{E}_{\pi_{\mathrm{tar}}}[\|\hat{s} - \nabla \log \pi_{\mathrm{tar}}\|^p]^{1/p}$ for any $p \geq 1$, while the law of Langevin dynamics run with $\hat{s}$ initialized at these $n$ samples remains far from $\pi_{\mathrm{tar}}$ for any $\mathrm{poly}(d)$ time horizon. Again, the TV distance between $\pi_{\mathrm{tar}}$ and the law of Langevin dynamics run with $\hat{s}$ from this initialization is $1 - e^{-\Omega(d)}$.

3. **Theorem 3.11, Subsection 3.3:** We prove further negative results for a wide range of target distributions and arbitrary initializations in the limit $t \to \infty$. We show for target distributions $\pi_{\mathrm{tar}}$ satisfying Assumption 3.10, as $t \to \infty$, the TV distance between $\pi_{\mathrm{tar}}$ and the law of Langevin dynamics run with $L^2$-accurate scores estimates is arbitrarily close to 1.

4. **In Section 4:** We also validate the practical prescription of Theorem 3.7 via simulation.

All the above results are for the continuous time Langevin *diffusion*, which is the *idealization* of discrete-time Langevin dynamics. As such we believe our result presents strong evidence against discrete-time Langevin dynamics as well. Moreover, as per Remark 3.3, 3.8, Theorems 3.1 and 3.7 directly generalize to the discrete-time Unadjusted Langevin Algorithm, a natural discretization of Langevin dynamics. Theorems 3.1 and 3.7 also generalize to strongly-log-concave targets; see Remarks 3.5 and 3.9.

**Significance of our results:** Our results demonstrate the Langevin dynamics is not robust to $L^2$ (more generally $L^p$) score estimation error. We emphasize that lack of robustness is the fundamental cause; the examples of Theorems 3.1, 3.7 are not degenerate. The target distribution is as simple as an isotropic Gaussian, the score estimates $\hat{s}$ are Lipschitz, and the initializations considered are natural.

Theorems 3.1, 3.11 provide 'worst-case' or 'adversarial' constructions that demonstrate $L^2$ (more generally $L^p$) score estimation error is in and of itself insufficient. This result stands in sharp contrast to diffusion models, where also convergence occurs in $\mathrm{poly}(d)$ time; as such our work provides further support for diffusion models from a novel angle.

*Our main result is Theorem 3.7*, which we believe carries further practical significance. In particular, this result prescribes one to use *fresh* samples not used in producing the estimate $\hat{s}$ when performing data-based initialization in practice. Moreover, this construction of $\hat{s}$ is – in some sense – based on $\hat{s}$ 'memorizing' all the $n$ i.i.d. samples. Such a situation has been empirically confirmed in certain settings in supervised learning with overparametrized neural networks,

as per e.g. Arpit et al. (2017); Zhang et al. (2017) – of which score matching is a prominent application. However, score functions are often trained on the entire dataset in practice. Our results serve to caution against doing so.

## 2. Background and Notation

**Notation:** We let $\pi_{\mathrm{tar}}$ denote the target distribution in $\mathbb{R}^d$. For a set $A$, we let $A^C$ denote its complement. We let $(B_t)_{t \geq 0}$ be a $d$-dimensional standard Brownian motion. For a random variable $X$ we let $\mathcal{L}(X)$ denote its law. We let $N(\theta, \Sigma)$ be a Gaussian with mean $\theta$ and covariance $\Sigma$. We let $\mathsf{TV}(P, Q)$ denote Total Variation (TV) distance between two probability distributions $P$ and $Q$. We let $\delta_x$ denote the Dirac Delta distribution at $x$. We let $\mathbb{1}$ denote the all ones vector. We write $a = \mathrm{poly}(b)$ for reals $a, b > 0$ if $a$ is upper bounded by a polynomial of fixed degree in $b$.

### 2.1. Langevin Dynamics

Letting $\pi_{\mathrm{tar}} \propto e^{-V(x)}$ be the target distribution on $\mathbb{R}^d$, the associated score function is $\nabla \log \pi_{\mathrm{tar}}(x) = -\nabla V(x)$. The overdamped Langevin dynamics is the stochastic differential equation (SDE)

$$\mathrm{d}X_t = \nabla \log \pi_{\mathrm{tar}}(X_t)\,\mathrm{d}t + \sqrt{2}\,\mathrm{d}B_t, \qquad X_0 \sim \mu_0, \quad (1)$$

where $\mu_0$ is arbitrary. Under mild regularity conditions, $\pi_{\mathrm{tar}}$ is the stationary distribution for the SDE (1), and $X_t \xrightarrow{d} \pi_{\mathrm{tar}}$ as $t \to \infty$ (Chiang & Chow, 1989).

**Non-Asymptotic Convergence:** The SDE (1) converges *efficiently* to $\pi_{\mathrm{tar}}$ in $\mathsf{TV}$ (in polynomial time in $d$) when $\pi_{\mathrm{tar}}$ satisfies a *Poincaré Inequality*, with stronger guarantees on the convergence when $\pi_{\mathrm{tar}}$ satisfies a *Log-Sobolev Inequality* (Villani et al., 2008; Bakry et al., 2014). The Poincaré Inequality holds for log-concave $\pi_{\mathrm{tar}}$ (convex $V(x)$) (Bobkov, 1999). Meanwhile the Log-Sobolev Inequality holds for strongly-log-concave $\pi_{\mathrm{tar}}$ (strongly convex $V(x)$) (Bakry & Émery, 2006). Specifically, if $\nabla^2 V(x) \succeq \alpha I_d$, then the Log-Sobolev constant of $\pi_{\mathrm{tar}}$ is at most $\frac{1}{\alpha}$.[4] In particular, *for well-conditioned Gaussians, e.g. isotropic Gaussians, we expect the SDE (1) to sample efficiently in $\mathsf{TV}$ from $\pi_{\mathrm{tar}}$.*

**Discretization:** One discretizes (1) to sample from $\pi_{\mathrm{tar}}$ with a discrete-time algorithm. There are several discretizations of (1) in the literature which enjoy polynomial convergence to $\pi_{\mathrm{tar}}$ in $\mathsf{TV}$ when $\pi_{\mathrm{tar}}$ is a Gaussian and more generally, when $V$ is strongly convex (Chewi, 2025), and when $\nabla V$ is Lipschitz (which is the case for our lower bound constructions in Theorems 3.1, 3.7). Perhaps the

---

[4]Or $\frac{2}{\alpha}$, depending on the normalization used.

most natural is the *Unadjusted Langevin Algorithm* (ULA):

$$X_{t+1} = X_t + \eta \nabla \log \pi_{\text{tar}}(X_t) + \sqrt{2\eta}\, \varepsilon_t, \qquad X_0 \sim \mu_0, \tag{2}$$

where $\eta > 0$ is the step size, $\mu_0$ is arbitrary, and $\varepsilon_t \sim N(0, I_d)$ are i.i.d. Gaussians. This algorithm has been widely studied as a natural discretization of (1), see e.g. Vempala & Wibisono (2019); Lee et al. (2022); Chewi et al. (2025). See Chewi (2025) for many more details. Note *all these discretizations aim to simulate the idealization* (1), see Chewi (2025) for further discussion.

## 2.2. Diffusion Models

Diffusion models were introduced in several key early works (Song & Ermon, 2019; Ho et al., 2020; Song et al., 2021a;b), and were originally motivated by the idea of *time-reversal*. One considers an SDE or other stochastic process that converges from $\pi_{\text{tar}}$ to $N(0, I_d)$, known as the *forward process*. One then applies the theory of time-reversal of SDEs (Haussmann & Pardoux, 1986) to obtain a stochastic process that converges from $N(0, I_d)$ to $\pi_{\text{tar}}$, known as the *reverse process*. One discretizes the reverse process, yielding an algorithm that samples from $\pi_{\text{tar}}$.

For a high-level overview, one frequently used choice of forward process, proposed in Song & Ermon (2019); Ho et al. (2020); Song et al. (2021b) (though certainly not the only one) is the Ornstein–Uhlenbeck (OU) process:

$$d\bar{X}_t = -\bar{X}_t\, dt + \sqrt{2}\, dB_t, \qquad \bar{X}_0 \sim \pi_{\text{tar}}. \tag{3}$$

Letting $\pi_t$ denote the law of $\bar{X}_t$, and fixing a terminal time $T \geq 0$, the corresponding reverse process is the SDE

$$dX_t = (X_t + 2\nabla \log \pi_{T-t}(X_t))\, dt + \sqrt{2}\, dB_t, \; X_0 \sim \pi_T. \tag{4}$$

By the theory of time-reversal of SDEs (Haussmann & Pardoux, 1986), the above reverse process converges to $\pi_{\text{tar}}$; moreover by convergence results for the OU process, for appropriate $T > 0$, $\pi_T \approx N(0, I_d)$. This is an example of the Denoising Diffusion Probabilistic Models (DDPM) framework (Ho et al., 2020). This is by no means the only type of algorithm subsumed by diffusion models. For example, one can also time-reverse the forward process (3) to yield an Ordinary Differential Equation (ODE); this is the Denoising Diffusion Implicit Models (DDIM) framework (Song et al., 2021a).

**Discretization:** To sample from $\pi_{\text{tar}}$ with a discrete-time algorithm, one discretizes (4) in the case of the DDPMs, and more generally discretizes the suitable reverse process (e.g. the ODE from DDIMs). This leads to a gradually noised sequence of distributions $\pi_0 \approx \pi_{\text{tar}}, \ldots, \pi_k$. Note

$\pi_0$ is only approximately equal to $\pi_{\text{tar}}$ due to early stopping often employed in practice, see e.g. Karras et al. (2022). To do so, one must learn estimates of the score functions $\nabla \log \pi_0, \ldots, \nabla \log \pi_k$ from i.i.d. data from $\pi_{\text{tar}}$ via *score matching*, see e.g. (Hyvärinen & Dayan, 2005; Vincent, 2011; Song & Ermon, 2019; Karras et al., 2022).

It has been established that with an estimated score $s$ and appropriate discretization, the sampling error in TV scales linearly with the $L^2$ score estimation error, within $\text{poly}(d)$ iterations. Specifically, the sampling error in TV scales linearly in a suitable weighted average of the $L^2$ score estimation errors of $\nabla \log \pi_0, \nabla \log \pi_1, \ldots, \nabla \log \pi_k$. This result applies for the reverse process of both DDPMs and DDIMs; see e.g. Chen et al. (2023b); Benton et al. (2024); Li et al. (2024); Jiao et al. (2025), among many other works.

## 3. Main Results

### 3.1. Standard normal initialization

Our first Theorem yields a lower bound for Langevin Dynamics when run with standard normal initialization.

**Theorem 3.1** (Standard Normal Initialization Lower Bound). *Consider $\pi_{\text{tar}} = N(\mu, I_d)$ for any vector $\mu$ with $\|\mu\| = 7\sqrt{d}$. For a universal constant $\alpha = \alpha_{A.6} > 0$ given in Lemma A.6, define the score estimate $\hat{s}(x)$ by*

$$\begin{cases} -\alpha x & : \|x\| \leq 4\sqrt{d} \\ -(x - \mu) & : \|x\| \geq 5\sqrt{d} \\ -\psi\left(\frac{x}{\sqrt{d}}\right) \cdot \alpha x - \left(1 - \psi\left(\frac{x}{\sqrt{d}}\right)\right) \cdot (x - \mu) & : else, \end{cases}$$

*where $\psi(\cdot) \in [0, 1]$ is a suitable interpolating bump function given in Lemma A.5. Then defining $X_t$ be the SDE*

$$dX_t = \hat{s}(X_t)\, dt + \sqrt{2}\, dB_t, \qquad X_0 \sim N(0, I_d), \tag{5}$$

*we have:*

1. *Small $L^p$ error: for any $p \geq 1$ and $d \geq d_0(p)$ large enough in terms of $p$,*

$$\underset{x \sim \pi_{\text{tar}}}{\mathbb{E}} \left[\|\hat{s}(x) - \nabla \log \pi_{\text{tar}}(x)\|^p\right]^{1/p} \leq e^{-\Omega(d)}. \tag{6}$$

2. *Far from $\pi_{\text{tar}}$ in Total Variation on sub-exponential time scales: for any $d \geq d_0$ where $d_0$ is a universal constant and any $T \leq e^{c_{A.6}d/2}$ where $c_{A.6}$ is a universal constant given in Lemma A.6,*

$$\mathsf{TV}\big(\mathcal{L}(X_T), \pi_{\text{tar}}\big) \geq 1 - e^{-\Omega(d)}. \tag{7}$$

*Furthermore, $\hat{s}$ is Lipschitz with constant independent of $d$.*

Since $\hat{s}$ is Lipschitz, by Theorem 5.2.1 of Oksendal (2013), the SDE (5) has a unique strong solution. We provide proof ideas next, and the full proof in Subsection B.1.

Theorem 3.1 establishes that in high dimensions, the law of (5) and $\pi_{\text{tar}}$ are very far away in TV in *any* polynomial time scales, despite $\hat{s}$ enjoying extremely small $L^p$ accuracy in high dimensions. However, this $\hat{s}$ is a adversarial or 'worst-case' construction. The failure mechanism for this Theorem arises due to initializing at $N(0, I_d)$, which has low density with respect to $\pi_{\text{tar}}$. In such regions, small $L^2$ error under $\pi_{\text{tar}}$ provides little control over $\hat{s}$. See the proof ideas for more details. We also note the choice 7 in $\|\mu\| = 7\sqrt{d}$ is arbitrary, and the desired effect occurs for other $\mu$; see the simulations supporting Theorem 3.1 in Section 4.

While Theorem 3.1 applies for dimensions $d \geq d_0(p)$ larger than a suitable universal constant, in our simulations in Section 4, we see effects (for Theorem 3.7, which uses similar results on high-dimensional concentration of measure) taking hold for $d = 50, 200$.[5]

**Corollary 3.2** (Mixing Time). *The conclusion of Theorem 3.1 also applies for any initialization $x_0$ with $\|x_0\| \leq 1.1\sqrt{d}$. Hence the* mixing time *of (5) to $\pi_{\text{tar}}$ is at least* $e^{c_{A.6}d/2}$.

*Remark* 3.3 (Discretization). Consider any $0 < T < e^{c_{A.6}d/2}$ and any discretization $\hat{X}_T$ of (5) such that $\mathsf{TV}\big(\mathcal{L}(\hat{X}_T), \mathcal{L}(X_T)\big) \leq c$. By Girsanov's Theorem, as $\hat{s}$ is Lipschitz, this applies for ULA initialized at $N(0, I_d)$ with suitable step size. See Chapter 4 of (Chewi, 2025). By the Triangle Inequality for TV we obtain the following lower bound for the discretization:

$$\mathsf{TV}\big(\mathcal{L}(\hat{X}_T), \pi_{\text{tar}}\big) \geq 1 - c - e^{-\Omega(d)}.$$

We note that closeness in TV likely applies for many discretizations of Langevin dynamics, as they aim to faithfully simulate the Langevin SDE (1) (Chewi, 2025).

**Proof ideas of Theorem 3.1:** We show the proof ideas for each of the claimed parts of the Theorem.

**Small $L^p$ error:** The $L^p$ error under $\pi_{\text{tar}}$ of $\|\hat{s} - \nabla \log \pi_{\text{tar}}\|$ is *exponentially small in high dimensions* by results on Gaussian concentration of the norm (see Lemma A.1). Under $\pi_{\text{tar}}$, the set $\big\{x \in \mathbb{R}^d : \|x\| \leq 5\sqrt{d}\big\}$ has exponentially small mass, while in this region, $\alpha\|\hat{s}(x) - \nabla \log \pi_{\text{tar}}(x)\|$ is at most polynomial in $d$. Essentially, in high dimensions one can 'hide' the bad set $\big\{x \in \mathbb{R}^d : \|x\| \leq 5\sqrt{d}\big\}$ w.r.t. $\pi_{\text{tar}}$.

**Lipschitz $\hat{s}$:** $\hat{s}$ is Lipschitz by direct calculation, as $\psi(\cdot)$ from Lemma A.5 has range in $[0, 1]$ and is Lipschitz with universal constant bound on its Lipschitz constant.

**Far from $\pi_{\text{tar}}$:** Suppose the initialization $x_0$ is such that $\|x_0\| \leq 1.1\sqrt{d}$; this is exponentially likely when we draw $x_0 \sim N(0, I_d)$ by results on Gaussian concentration of the

norm (Lemma A.1).

If $\|x_0\| \leq 1.1\sqrt{d}$, before reaching $A := \big\{x \in \mathbb{R}^d : \|x\| \geq 4\sqrt{d}\big\}$, we have $\hat{s}(x) = -\alpha x$. Thus the escape time of (5) from $A^C$ and the escape time of the following SDE are identical in distribution:

$$\mathrm{d}\bar{X}_t = -\alpha\bar{X}_t\mathrm{d}t + \sqrt{2}\mathrm{d}B_t, \qquad \bar{X}_0 = x_0. \tag{8}$$

The SDE (8) is a rescaled Ornstein-Uhlenbeck (OU) process with stationary distribution $N(0, \frac{1}{\alpha}I_d)$. Since $\alpha$ is a large enough universal constant, for initializations $\|x_0\| \leq 1.1\sqrt{d}$ well within $A^C$, the escape time of (8) from $A^C$ is exponential in $d$ with very high probability. Specifically, letting $X_t(x_0)$ denote the law of (5) when initialized at $x_0$, we have the following Lemma.

**Lemma 3.4.** *Suppose $d \geq K_{A.6}$ for a universal constant $K_{A.6} > 0$. Consider any $T \leq e^{c_{A.6}d/2}$. Let $\tau(x_0) = \inf_{t \geq 0} \big\{\|X_t(x_0)\| \geq 4\sqrt{d}\big\}$. If $\|x_0\| \leq 1.1\sqrt{d}$, then with probability at least $1 - 3\alpha e^{-c_{A.6}d}$, $\tau(x_0) > T$.*

See Subsection B.1 for the proof. Now to establish the desired lower bound on $\mathsf{TV}\big(\mathcal{L}(X_T), \pi_{\text{tar}}\big)$, we use that $A^C$ has exponentially small mass under $\pi_{\text{tar}}$ by Gaussian concentration (Lemma A.1), while $X_t$ only reaches $A$ with exponentially small probability within $[0, T]$ by Lemma 3.4. Thus, $\pi_{\text{tar}}$ puts almost all its mass on $A$, while $\mathcal{L}(X_T)$ puts almost no mass on it, yielding the desired lower bound on TV distance. The full proof is in Subsection B.1.

*Remark* 3.5 (Generalizing $\pi_{\text{tar}}$ in Theorem 3.1). Theorem 3.1 generalizes to when $\pi_{\text{tar}} \propto e^{-V(x)}$ is strongly log-concave (when $V(x)$ is strongly convex) and when $\nabla V$ is Lipschitz. We define $\hat{s}$ analogously to Theorem 3.1, except each instantiation of $-(x - \mu)$ is replaced with $-\nabla V$. The escape time of (5) from $\big\{x : \|x\| \leq 4\sqrt{d}\big\}$ still equals that of (8), as here $\hat{s}$ is still $-\alpha x$ in this region. The proof then proceeds identically as the proof of Theorem 3.1 discussed above. The one difference is to establish that the $L^p$ error of $\|\hat{s} - \nabla \log \pi_{\text{tar}}\|$ with respect to $\pi_{\text{tar}}$ is small. Here rather than concentration of the Gaussian measure, we apply concentration of Lipschitz functions of strongly log-concave functions (Bakry et al., 2014).

### 3.2. Data-based initialization

A natural strategy for initializing Langevin dynamics when one has a score estimate $\hat{s}$ learned from data drawn from $\pi_{\text{tar}}$ is *data-based initialization*, studied in Koehler & Vuong (2024); Koehler et al. (2025). Here, one initializes Langevin dynamics from i.i.d. samples $x_1, \ldots, x_n$ drawn from $\pi_{\text{tar}}$: we let $X_0 \sim \frac{1}{n}\sum_{i=1}^{n} \delta_{x_i}$, the empirical distribution of these samples. This is a natural initialization when one has i.i.d. samples from $\pi_{\text{tar}}$ and aims to produce more of them. As discussed in Koehler & Vuong (2024), p. 4, data-based initialization is also closely related to contrastive divergence

---

[5]One can readily find a bound on $d_0$, $d_0(p)$ by tracking the proof; we did not pursue this for notational simplicity.

(Hinton, 2002; 2012; Xie et al., 2016; Gao et al., 2018).

Koehler & Vuong (2024); Koehler et al. (2025) established several appealing properties of data-based initialization:

1) It is robust to $L^2$-error (with respect to $\pi_{\text{tar}}$) in the score estimates $\hat{s}$, a natural assumption when $\hat{s}$ is learned from data drawn from $\pi_{\text{tar}}$.

2) With $n = \text{poly}(d)$ samples, it can alleviate the exponential in $d$ mixing time of Langevin dynamics from many mixture distributions (Bovier et al., 2004; 2005; Menz & Schlichting, 2014) (though Langevin dynamics mixes rapidly for $\pi_{\text{tar}}$ in Theorem 3.7).

Our result, Theorem 3.7, complements the work of Koehler & Vuong (2024); Koehler et al. (2025), by showing that the robustness of data-based initialization only applies when *fresh* samples different from those used to learn $\hat{s}$ are used for the initialization. We believe this serves as an important warning to use *fresh* samples when applying data-based initialization or similar methods in practice. This is the main result of this paper.

To state Theorem 3.7, we need the following definition.

**Definition 3.6** (General Position). We say $x_1, \ldots, x_n \in \mathbb{R}^d$ are in *general position* if $\|x_i - x_j\| \geq 0.4\sqrt{d}$ for all $1 \leq i \neq j \leq n$ and $0.5\sqrt{d} \leq \|x_i\| \leq 2\sqrt{d}$ for all $1 \leq i \leq n$.

Note by Triangle Inequality that if $x_1, \ldots, x_n$ are in general position, for all $x \in \mathbb{R}^d$, exactly one of the following holds: 1) $\|x - x_i\| \geq 0.16\sqrt{d}$ for all $1 \leq i \leq n$, or 2) there is a *unique* $i, 1 \leq i \leq n$ such that $\|x - x_i\| < 0.16\sqrt{d}$.

**Theorem 3.7** (Data-Based Initialization Lower Bound). *Consider* $\pi_{\text{tar}} = N(0, I_d)$. *Consider* $n = \text{poly}(d)$ *samples* $x_1, \ldots, x_n \in \mathbb{R}^d$ *drawn i.i.d. from* $\pi_{\text{tar}}$. *Then* $x_1, \ldots, x_n$ *are in general position with probability at least* $1 - e^{-\Omega(d)}$ *for* $d \geq d_0$ *where* $d_0$ *is a universal constant, and if* $x_1, \ldots, x_n$ *are in general position the following holds. For a universal constant* $\alpha = \alpha_{A.6} > 0$ *given in Lemma A.6, define the score estimate* $\hat{s}(x)$ *as follows. Let* $\psi(\cdot) \in [0, 1]$ *be a suitable interpolating bump function given in Lemma A.5. Now:*

1. *If* $\|x - x_i\| \geq 0.16\sqrt{d}$ *for all* $1 \leq i \leq n$, $\hat{s}(x) = -x$.

2. *Otherwise, we let* $i \in [n]$ *be the* unique *index such that* $\|x - x_i\| < 0.16\sqrt{d}$. *If* $\|x - x_i\| \leq 0.15\sqrt{d}$, *set* $\hat{s}(x) = -\alpha(x - x_i)$. *Else if* $0.15\sqrt{d} < \|x - x_i\| < 0.16\sqrt{d}$, *set* $\hat{s}(x) = -\psi\left(\frac{100(x - x_i) - 11}{\sqrt{d}}\right) \cdot \alpha(x - x_i) - \left(1 - \psi\left(\frac{100(x - x_i) - 11}{\sqrt{d}}\right)\right) \cdot x$.

*Then defining* $X_t$ *be the SDE*

$$\mathrm{d}X_t = \hat{s}(X_t)\,\mathrm{d}t + \sqrt{2}\,\mathrm{d}B_t, \qquad X_0 \sim \frac{1}{n}\sum_{i=1}^{n}\delta_{x_i}, \quad (9)$$

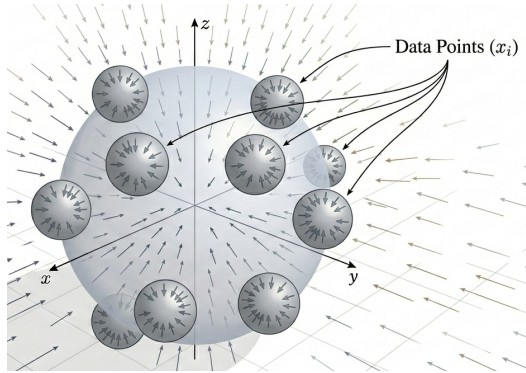

*Figure 1.* A schematic of the construction in Theorem 3.7. For each ball centered around each $x_i$, the gradient field is perturbed to point towards the center of the ball.

*we have:*

1. *Small* $L^p$ *error: for any* $p \geq 1$ *and* $d \geq d_0(p)$ *large enough in terms of* $p$,

$$\mathbb{E}_{x \sim \pi_{\text{tar}}}[\|\hat{s}(x) - \nabla \log \pi_{\text{tar}}(x)\|^p]^{1/p} \leq e^{-\Omega(d)}. \quad (10)$$

2. *Far from* $\pi_{\text{tar}}$ *in Total Variation on sub-exponential time scales: for* $d \geq d_0$ *where* $d_0$ *is a universal constant and any* $T \leq e^{c_{A.6}d/2}$ *where* $c_{A.6}$ *is a universal constant given in Lemma A.6,*

$$\mathsf{TV}\big(\mathcal{L}(X_T), \pi_{\text{tar}}\big) \geq 1 - e^{-\Omega(d)}. \quad (11)$$

*Finally,* $\hat{s}$ *is Lipschitz with constant independent of* $d$.

Note the construction of $\hat{s}$ is justified by the discussion in Definition 3.6. Again since $\hat{s}$ is Lipschitz, by Theorem 5.2.1 of Oksendal (2013), the SDE (5) has a unique strong solution. We prove Theorem 3.7 in Subsection B.2; at a high level, the proof is similar to that of Theorem 3.1. A schematic of the construction in Theorem 3.7 is given in Figure 1. Finally, while Theorem 3.7 applies for dimensions $d \geq d_0(p)$ large enough, in our simulations in Section 4, we see the claimed effects taking hold for $d = 50, 200$.

Note the construction of $\hat{s}$ in Theorem 3.7 is based on $\hat{s}$ 'memorizing' the score function of $N(x_i, \frac{1}{\alpha}I_d)$ for each $x_i$. One can view $\hat{s}$ as – in some sense – having memorized the training samples. This is a natural situation to expect in practice when performing supervised learning with over-parametrized neural networks – of which score matching is a prominent application – as per e.g. Arpit et al. (2017); Zhang et al. (2017).

In our simulations in Section 4, we show such situations can occur when estimating $\nabla \log \pi_{\text{tar}}$ with an overparametrized neural network, for $\pi_{\text{tar}}$ as simple as a Gaussian. There, we also show that Langevin dynamics run with the score

estimate $\hat{s}$ initialized at fresh samples performs significantly better than when initialized at the samples used to learn $\hat{s}$. As such, our simulations validate the prescription from Theorem 3.7.

*Remark* 3.8 (Discretization). Analogously to Theorem 3.1, Theorem 3.7 generalizes to any discretization $\hat{X}_T$ of (9) that is TV-close to (9), such as ULA initialized from $\frac{1}{n}\sum_{i=1}^{n}\delta_{x_i}$ with suitable step size.

*Remark* 3.9 (Generalizing $\pi_{\text{tar}}$ in Theorem 3.7). As with Theorem 3.1, Theorem 3.7 also holds for strongly-log-concave $\pi_{\text{tar}}$. This is because for $n = \text{poly}(d)$, $x_1, \ldots, x_n \overset{\text{i.i.d.}}{\sim} \pi_{\text{tar}}$ are in general position with probability at least $1 - e^{-\Omega(d)}$ for $d \geq d_0$, where $d_0$ is a universal constant. From here, one can follow the exact same proof as that of Theorem 3.7. Specifically, for our concentration results we now use concentration of Lipschitz functions of strongly-log-concave measures (Bakry et al., 2014).

### 3.3. Lower bounds for general target distributions

In this section, we consider a broad class of target distributions $\pi_{\text{tar}}$ beyond the Gaussian case. For *all* such distributions and *all* initializations, we establish a lower bound on the efficacy of Langevin dynamics in the asymptotic limit $t \to \infty$. We consider a target distribution $\pi_{\text{tar}}$ on $\mathbb{R}^d$ that satisfies the following assumptions.

**Assumption 3.10.** Assume that (1) $\nabla \log \pi_{\text{tar}}$ exists almost everywhere and is Lipschitz continuous, (2) $\mathbb{E}_{\pi_{\text{tar}}}[\|\nabla \log \pi_{\text{tar}}(x)\|^2] < \infty$, (3) the density of $\pi_{\text{tar}}$ is strictly positive w.r.t. Lebesgue measure, and (4) $\log \pi_{\text{tar}}$ satisfies the dissipativity assumption of Raginsky et al. (2017).

We show in this section that for any $\varepsilon_{\text{score}}, \varepsilon_{\text{TV}} > 0$, there exists a score estimate $\hat{s}$ with small $L^2$ error but such that Langevin Dynamics has large TV sampling error. Here $\hat{s}$ is a 'worst-case' construction. Specifically, we establish:

**Theorem 3.11.** *For any $\varepsilon_{\text{score}}, \varepsilon_{\text{TV}} > 0$, there exist a unit vector $u \in \mathbb{R}^d$ and $\theta \in (0, \pi/2)$ such that the following holds. There exists a score estimate $\hat{s}$ that is piecewise Lipschitz such that we have:*

1. *Small $L^2$ error:*

$$\mathbb{E}_{\pi_{\text{tar}}}[\|\hat{s}(x) - \nabla \log \pi_{\text{tar}}(x)\|^2] \leq \varepsilon_{\text{score}}^2.$$

2. *Far from $\pi_{\text{tar}}$ in Total Variation as $t \to \infty$: for any initial distribution $\pi_0$, define the SDE*

$$\mathrm{d}X_t = \hat{s}(X_t)\mathrm{d}t + \sqrt{2}\mathrm{d}B_t, \qquad X_0 \sim \pi_0,$$

*and let $\pi_t$ be the distribution of $X_t$. Then*

$$\liminf_{t \to \infty} \text{TV}(\pi_{\text{tar}}, \pi_t) \geq 1 - \varepsilon_{\text{TV}}.$$

Note the SDE given above has a unique strong solution by Zvonkin's Theorem (Zvonkin, 1974) as $\hat{s}$ is piecewise Lipschitz. We prove Theorem 3.11 in Appendix C.

Theorem 3.11 establishes a lower bound against the efficacy of Langevin dynamics run with a score estimate with small $L^2$ error for a broad class of target distributions $\pi_{\text{tar}}$ and *any* initialization, that holds in the limit $t \to \infty$.

## 4. Simulations

We validate the practical implication of Theorem 3.7 via simulation. We emphasize that the simulations here are intended only to provide support for Theorem 3.7; an additional simulation supporting Theorem 3.1 is provided at the end of this Section. Specifically, we consider two target distributions $\pi_{\text{tar}}$: a single Gaussian $N(\mathbb{1}, 2I_d)$ with $d = 50$ and a mixture of Gaussians (GMM) $\frac{1}{2}N(-\mathbb{1}, 2I_d) + \frac{1}{2}N(4\mathbb{1}, 2I_d)$ with $d = 25$. Full details and results are in Section D.

**Learning the score:** We learn an estimate $\hat{s}$ of the score function $\nabla \log \pi_{\text{tar}}$ with a fully connected neural network with 3 hidden layers trained with Adam (Kingma & Ba, 2015) for 150,000 epochs. To encourage an overfit or 'memorized' score estimate $\hat{s}$, we create a training set of 10,000 samples by drawing 1000 i.i.d. samples from $\pi_{\text{tar}}$ and duplicating each sample 10 times. We learn the score by score matching; we learn the denoiser at the 100 lowest noise levels out of a linear noise schedule of 1000 levels (e.g. Ho et al. (2020)). We intentionally used a limited dataset of 1000 repeated points to encourage overfitting of the score function to these training points, and therefore demonstrate the phenomenon predicted by Theorem 3.7.

Here, we used the DDPM objective with low noise levels because it yielded the most accurate and stable approximation of $\nabla \log \pi_{\text{tar}}$. Using a single, very low noise level proved difficult to train reliably. While methods such as implicit score matching or sliced score matching could in principle be used, they are less practical even at small scale. We chose DDPM with low noise levels because it is fast, easy to implement, and sufficient to illustrate the phenomenon highlighted by Theorem 3.7. During training, we added noise 'dynamically': for each batch, new noise was sampled and added to the base samples from $\pi_{\text{tar}}$, following the standard DDPM procedure.

**Sampling algorithms:** We run fixed-step-size Langevin dynamics for 1000 iterations, evaluating the denoiser at the 10th lowest noise level out of the noise schedule to approximate $\nabla \log \pi_{\text{tar}}$ at each iteration. We use 1000 iterations to simulate a $\text{poly}(d)$ timescale, and as score-based generative models rarely use more than 1000 denoiser evaluations in practice (Karras et al., 2022). We initialize Langevin dynamics at $n$ points to produce $n \in \{1500, 7500, 13500, 16250\}$ samples the following three ways, yielding 3 different al-

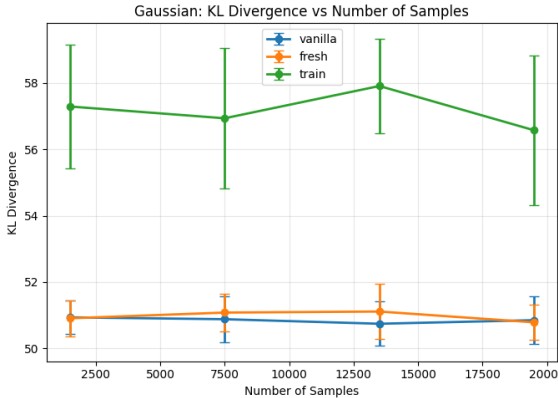

*Figure 2.* KL of produced samples vs Gaussian $\pi_{\text{tar}}$. KL is estimated by fitting a Gaussian $\hat{\pi}$ to the produced samples and analytically computing $\mathsf{KL}(\pi_{\text{tar}}, \hat{\pi})$.

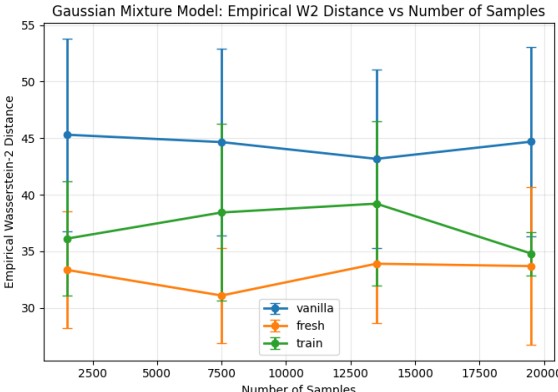

*Figure 3.* Empirical Wasserstein distance of produced samples vs GMM $\pi_{\text{tar}}$. Wasserstein distance was approximated via the Sinkhorn loss (Cuturi, 2013).

gorithms. 1) initialize $n$ draws from $N(0, I_d)$, 2) take 30 fresh i.i.d. draws from $\pi_{\text{tar}}$ each duplicated $n/30$ times, and 3) take 30 random draws from the 1000 distinct training samples each duplicated $n/30$ times. Here the 30 samples represent a random subset of the training samples or fresh samples; we repeat $n/30$ times to produce $n$ total samples. This whole procedure is done for 10 trials, and values are averaged over the 10 trials.

**Takeaway:** We plot our results in Figure 2 for Gaussian $\pi_{\text{tar}}$ and Figure 3 for GMM $\pi_{\text{tar}}$. Algorithms 1, 2, 3 are denoted 'vanilla', 'fresh', 'train' respectively. As predicted by Theorem 3.7, initializing from samples drawn from the training set (Algorithm 3) generally produces worse results than when the samples are fresh (Algorithm 2). For Gaussian $\pi_{\text{tar}}$, the gap between Algorithms 2 and 3 is significant, and Algorithm 1 performs similarly to Algorithm 2. For GMM $\pi_{\text{tar}}$, the gap between Algorithms 2 and 3 is smaller, although Algorithm 3 still generally does worse than Algo-

*Table 1.* Simulating the setting of Theorem 3.1 with target $\pi_{\text{tar}} = N(\mu, I_d)$, $d = 50$, and $\alpha = 25$.

| $\|\mu\|$ | $L_2$ score error | KL with $\hat{s}$ | KL with true score $s$ |
|---|---|---|---|
| 7.07 | $2.45 \times 10^2$ | $1.13 \times 10^3$ | 0.308 |
| 14.1 | $3.92 \times 10^2$ | $2.98 \times 10^3$ | 0.815 |
| 21.2 | $5.56 \times 10^2$ | $6.07 \times 10^3$ | 1.66 |
| 28.3 | $7.00 \times 10^2$ | $1.04 \times 10^4$ | 2.84 |
| 35.4 | 9.22 | $1.60 \times 10^4$ | 4.35 |
| 42.4 | 0 | $2.28 \times 10^4$ | 6.20 |
| 49.5 | 0 | $3.08 \times 10^4$ | 8.39 |

rithm 2, and Algorithm 1 performs the worst of the three methods due to the poor spectral gap of the GMM.

Note our results are consistent with those of (Koehler & Vuong, 2024), (Koehler et al., 2025) as Algorithm 3 is initialized at the exact same training samples used to learn $\hat{s}$. In contrast, the sampling results of (Koehler & Vuong, 2024), (Koehler et al., 2025) apply when initializing Langevin Dynamics at *fresh* samples.

Finally, we do not claim that the simulations verify every assumption of Theorem 3.7 exactly; the goal of these simulations is not a literal instantiation of the Theorem's assumptions. Rather, the simulations are designed to test its qualitative prediction in synthetic settings: when the score estimate is overfit to the training samples, Langevin dynamics can fail when initialized from these same training samples.

**Simulations in support of Theorem 3.1** We run Langevin dynamics for 1000 iterations with the analytic form of $\hat{s}$ from Theorem 3.1, and with the true score $s$. This is done for $\pi_{\text{tar}} = N(\mu, I_d)$ where $\|\mu\| = k\sqrt{d}$, $k \in \{1, \dots, 7\}$, $d = 50$, and setting $\alpha = 25$ in the construction of $\hat{s}$ in Theorem 3.1. We perform this for 10 trials and plot the mean $L_2$ score error of $\hat{s}$, and the mean KL divergence of the samples produced by running Langevin with $\hat{s}$ and with $s$. See Table 1.

The claimed effect in Theorem 3.1 takes places when $\|\mu\| = k\sqrt{d}$ for $k = 6, 7$, where $L^2$ score error of $\hat{s}$ is extremely small (it is 0 to all the significant digits), but the KL divergence of the samples produced by running Langevin with $\hat{s}$ is much larger than when running Langevin with $s$. Note when $k = 6, 7$, the ball of radius $\sqrt{d}$ centered at $\mu$ is completely contained in $\{x : \|x\| \geq 5\sqrt{d}\}$, corresponding to the setting of Theorem 3.1.

## 5. Conclusion

In this paper, we established that Langevin dynamics is not robust to $L^p$ errors in the estimate of the score function in high dimensions, even when the $L^p$ score estimation

error is exponentially small. Our lower bounds apply to natural examples: simple isotropic Gaussian target distributions, Lipschitz score estimates, and natural initializations, namely from $N(0, I_d)$ and from i.i.d. samples used to learn the score. As such, our work cautions against the use of Langevin dynamics with an estimated score. That being said, we emphasize that our constructions from Theorems 3.1, 3.11 are adversarial constructions that demonstrate $L^p$ score estimation error is by and itself insufficient. Understanding whether such constructions can arise when learning the score in practice is an interesting future direction.

## Impact Statement

This paper presents work whose goal is to advance the field of Machine Learning. There are many potential societal consequences of our work, none which we feel must be specifically highlighted here.

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

## A. Technical Preliminaries

**Additional Notation:** We let $\mathcal{S}^{d-1}$ denote the unit sphere in $\mathbb{R}^d$.

We first introduce several key technical preliminaries. The first two results we present are on concentration of measure in high dimensions.

**Lemma A.1** (Gaussian concentration of norm; see e.g. Theorem 3.1.1 and (3.7), Vershynin (2018)). *Let $X \sim N(0, I_d)$. There exist absolute constants $c, C > 0$ such that for all $t > 0$,*

$$\mathbb{P}\Big(\big|\|X\| - \sqrt{d}\big| \geq t\sqrt{d}\Big) \leq Ce^{-ct^2 d}.$$

**Lemma A.2** (Gaussian concentration of Lipschitz functions; see e.g. (5.4.2), Bakry et al. (2014)). *Suppose $f$ is a $L$-Lipschitz function. Then there exists an absolute constant $C > 0$ such that for all $t > 0$,*

$$\mathbb{P}_{X \sim N(0, I_d)}\Big(\big|f(X) - \mathbb{E}_{X \sim N(0, I_d)}[f(X)]\big| \geq t\Big) \leq 2e^{-\frac{t^2}{2CL^2}}.$$

Solely for the proof of Lemma A.7, we will need the following more involved results on concentration of measure to control a suitable stochastic process indexed by a continuous time index. The first result is Dudley's chaining bound, used to control the expectation of a relevant supremum.

**Theorem A.3** (Dudley Chaining; see e.g. Theorem 8.1.3 and Remark 8.1.9, Vershynin (2018)). *Let $(X_t)_{t \in T}$ be a mean 0 Gaussian process on a set $T$. Define a metric on $T$ by $d(t_1, t_2) = \mathbb{E}\left[(X_{t_1} - X_{t_2})^2\right]^{1/2}$. Then*

$$\mathbb{E}\Big[\sup_{t \in T} X_t\Big] \leq C \int_0^{diam(T)} \sqrt{\log \mathcal{N}(T, d, \varepsilon)} d\varepsilon,$$

*where $diam(T)$ denotes the diameter of $T$ w.r.t. $d(\cdot, \cdot)$, where $\mathcal{N}(T, d, \varepsilon)$ denotes the covering number of $T$ to scale $\varepsilon$ with respect to $d(\cdot, \cdot)$, and where $C > 0$ is an absolute constant.*

We will also use the Borell-TIS (Tsirelson–Ibragimov–Sudakov) Inequality to obtain concentration of a relevant supremum.

**Theorem A.4** (Borell-TIS Inequality; see e.g. Theorem 2.1.1, Adler & Taylor (2007)). *Let $f_t$ be a centered Gaussian process almost surely bounded on $T$. Then $\mathbb{E}\left[\sup_{t \in T} f_t\right] < \infty$ and letting $\sigma_T^2 := \sup_{t \in T} \mathbb{E}\left[f_t^2\right]$, we have*

$$\mathbb{P}\Big(\sup_{t \in T} f_t - \mathbb{E}\Big[\sup_{t \in T} f_t\Big] > u\Big) \leq e^{-\frac{u^2}{2\sigma_T^2}}.$$

Finally, we will use the following Lemma to construct the bump function $\chi(\cdot)$, which allows us to take $\hat{s}$ Lipschitz in Theorems 3.1 and 3.7.

**Lemma A.5** (Construction of bump function; Lemma 47, Chen & Sridharan (2025)). *We can construct a bump function $\chi(\cdot) \in [0, 1]$ such that:*

- *$\chi \equiv 0$ in $\{x : \|x\| \leq 4\}$, $\chi \equiv 1$ in $\{x : \|x\| \geq 5\}$.*

- *$\chi$ is differentiable to all orders.*

- *$\|\nabla \chi\|, \|\nabla^2 \chi\|_{op} \leq H$ for a universal constant $H > 0$.*

### A.1. Proof of Lemma A.6

We will need the following key technical Lemma to prove Theorems 3.1 and 3.7.

**Lemma A.6.** *There exists a universal constant $c_{A.6} > 0$ such that the following holds. Consider any $d \geq K_{A.6}$ where $K_{A.6} > 0$ is a universal constant and any $T \leq e^{c_{A.6} d/2}$. For any $\alpha \geq \alpha_{A.6}$ where $\alpha_{A.6} \geq 1$ is a universal constant,*

$$\mathbb{P}\left(\sup_{t \in [0,T]} \Big\| \int_0^t e^{-\alpha(t-s)} dB_s \Big\| > 0.1\sqrt{d}\right) \leq 3\alpha e^{-c_{A.6} d}.$$

To prove Lemma A.6, we will first need the following Lemma.

**Lemma A.7.** *Let $B_t$ be a standard Brownian motion in $\mathbb{R}^d$. Then there exists universal constants $K_{A.7}, c_{A.7} > 0$ such that*

$$\mathbb{P}\left(\sup_{t \in [0,1]} \left\| \int_0^t e^s \mathrm{d}\tilde{B}_s \right\| > K_{A.7}\sqrt{d}\right) \le e^{-c_{A.7}d}.$$

*Proof of Lemma A.7.* Note we can write $\left\| \int_0^t e^s \mathrm{d}B_s \right\| = \sup_{v \in \mathcal{S}^{d-1}} \left\langle \int_0^t e^s \mathrm{d}B_s, v \right\rangle$, where $\mathcal{S}^{d-1}$ denotes the unit sphere in $\mathbb{R}^d$. Thus $\mathbb{E}\left[\sup_{t \in [0,1]} \left\| \int_0^t e^s \mathrm{d}B_s \right\|\right] = \mathbb{E}\left[\sup_{t \in [0,1], v \in \mathcal{S}^{d-1}} \left\langle \int_0^t e^s \mathrm{d}B_s, v \right\rangle\right]$. Let $\bar{Z}_{t,v} := \left\langle \int_0^t e^s \mathrm{d}B_s, v \right\rangle = \int_0^t e^s \langle v, \mathrm{d}B_s \rangle$. Note the $\bar{Z}_{t,v}$ is a centered Gaussian process.

Now we use Dudley Chaining (Theorem A.3) to upper bound $\mathbb{E}\left[\sup_{t \in [0,1]} \left\| \int_0^t e^s \mathrm{d}B_s \right\|\right] = \mathbb{E}\left[\sup_{t \in [0,1], v \in \mathcal{S}^{d-1}} \bar{Z}_{t,v}\right]$, with $T = [0,1] \times \mathcal{S}^{d-1}$. Recall the metric is $d\big((v_1, t_1), (v_2, t_2)\big) = \mathbb{E}\left[(\bar{Z}_{t_1,v_1} - \bar{Z}_{t_2,v_2})^2\right]^{1/2}$ for $0 \le t_1 \ne t_2 \le 1$, $v_1, v_2 \in \mathcal{S}^{d-1}$. Suppose WLOG that $t_1 \le t_2$. Note that for a vector $u$, $\langle u, \mathrm{d}B_s \rangle$ is a 1d Brownian motion scaled by $\|u\|$. We compute via Itô's Isometry (see e.g. Theorem 6.1, Steele (2001)),

$$\begin{aligned}
\mathbb{E}\left[(\bar{Z}_{t_1,v_1} - \bar{Z}_{t_2,v_2})^2\right] &= \mathbb{E}\left[\left(\int_0^{t_1} e^s \langle v_1, dB_s \rangle - \int_0^{t_2} e^s \langle v_2, dB_s \rangle\right)^2\right] \\
&= \mathbb{E}\left[\left(\int_0^{t_1} e^s \langle v_1 - v_2, dB_s \rangle - \int_{t_1}^{t_2} e^s \langle v_2, dB_s \rangle\right)^2\right] \\
&\le 2\,\mathbb{E}\left[\left(\int_0^{t_1} e^s \langle v_1 - v_2, dB_s \rangle\right)^2 + \left(\int_{t_1}^{t_2} e^s \langle v_2, dB_s \rangle\right)^2\right] \\
&= 2\|v_1 - v_2\|^2 \int_0^1 e^{2s} \mathrm{d}s + 2\|v_2\|^2 \int_{t_1}^{t_2} e^{2s} \mathrm{d}s \\
&\le 8\|v_1 - v_2\|^2 + 8(t_2 - t_1)\,.
\end{aligned}$$

A similar argument applies when $t_1 > t_2$, and consequently we have

$$d\big((v_1, t_1), (v_2, t_2)\big) \le \sqrt{8\|v_1 - v_2\|^2 + 8|t_2 - t_1|} \le 3\big(\|v_1 - v_2\| + |t_2 - t_1|^{1/2}\big)$$

and $\mathrm{diam}(T) \le 7$. Moreover, this shows that the covering number $\mathcal{N}(T, d, \varepsilon)$ of $T$ w.r.t. $d(\cdot, \cdot)$ is at most $N([0,1], d_1, \varepsilon/6) \cdot \mathcal{N}(\mathcal{S}^{d-1}, d_2, \varepsilon/6)$, where $d_1(v_1, v_2) = \|v_1 - v_2\|$ and $d_2(t_1, t_2) = |t_1 - t_2|^{1/2}$. Hence, $\mathcal{N}(T, d, \varepsilon) \le N([0,1], d_1, \varepsilon/6) \cdot \mathcal{N}(\mathcal{S}^{d-1}, d_2, \varepsilon/6) \le \left(\frac{C}{\varepsilon}\right)^d \frac{C}{\varepsilon^2} \le \left(\frac{C}{\varepsilon}\right)^{3d}$, where $C > 0$ is an absolute constant. Consequently again letting $C > 0$ denote a large enough absolute constant, Dudley Chaining (Theorem A.3) yields

$$\begin{aligned}
\mathbb{E}\left[\sup_{t \in [0,1]} \left\| \int_0^t e^s \mathrm{d}B_s \right\|\right] = \mathbb{E}\left[\sup_{t \in [0,1], v \in \mathcal{S}^{d-1}} \left\langle \int_0^t e^s \mathrm{d}B_s, v \right\rangle\right] \\
\le C \int_0^7 \sqrt{3d \log(C/\varepsilon)}\,\mathrm{d}\varepsilon \\
= C\sqrt{d} \int_0^7 \sqrt{\log(C) + \log(1/\varepsilon)}\,\mathrm{d}\varepsilon \\
\le K'_{A.7}\sqrt{d}, \quad (12)
\end{aligned}$$

by defining $K'_{A.7} > 0$ as a large enough universal constant.

Next we establish a suitable tail bound. We apply the Borell-TIS Inequality (Theorem A.4) to a suitable Gaussian process. Recalling that $\bar{Z}_{t,v} := \left\langle \int_0^t e^s \mathrm{d}B_s, v \right\rangle = \int_0^t e^s \langle v, \mathrm{d}B_s \rangle$, applying Itô's Isometry and using $0 \le t \le 1$ yields

$$\mathbb{E}\left[\bar{Z}_{t,v}^2\right] = \int_0^t e^{2s}\|v\|^2 \mathrm{d}t = \frac{1}{2}(e^{2t} - 1) \le 4.$$

Again, note the $\bar{Z}_{t,v}$ is a centered Gaussian process. Moreover $[0,1] \times \mathcal{S}^{d-1}$ is compact and $ve^s$ is continuous in $s$. Since the integrand is square integrable, it follows that $\bar{Z}_{t,v}$ is almost-surely bounded. Now the Borell-TIS Inequality applied to the $\bar{Z}_{t,v}$ gives

$$\mathbb{P}\left(\sup_{t \in [0,1], v \in \mathcal{S}^{d-1}} \bar{Z}_{t,v} \geq \mathbb{E}\left[\sup_{t \in [0,1], v \in \mathcal{S}^{d-1}} \bar{Z}_{t,v}\right] + t'\right) \leq e^{-t'^2/8}. \tag{13}$$

Noting $\sup_{t \in [0,1], v \in \mathcal{S}^{d-1}} \bar{Z}_{t,v} = \sup_{t \in [0,1]} Z_t$ and taking $t' = \sqrt{d}$ in the above, we conclude the proof upon taking $c_{A.7} = \frac{1}{8}$, $K_{A.7} = K'_{A.7} + 1$, and combining (12) and (13). $\qquad\square$

Now we can prove Lemma A.6. Since $\sup_{t \in [0,T]} \left\| \int_0^t e^{-\alpha(t-s)} dB_s \right\|$ does not decrease with $T$, by increasing $T$ to the nearest integer multiple of $\frac{1}{\alpha}$, we may without loss of generality suppose $T\alpha$ is an integer that is at least 1. Let $Z_t := \int_0^t e^{-\alpha(t-s)} dB_s$. Let $E = \left\{ \max_{t \in \{0,1,\ldots,T\alpha\}} \|Z_{\frac{t}{\alpha}}\| > 0.03\sqrt{d} \right\}$. Note the desired event can be written as

$$\left\{ \sup_{t \in [0,T]} \|Z_t\| > 0.1\sqrt{d} \right\} \subseteq E \cup \left( E^C \cap \bigcup_{t=0}^{T\alpha-1} \left\{ \sup_{s \in [\frac{t}{\alpha}, \frac{t+1}{\alpha}]} \|Z_s - Z_{\frac{t}{\alpha}}\| > 0.07\sqrt{d} \right\} \right). \tag{14}$$

We first upper bound $\mathbb{P}(E)$. A direct calculation shows that for any $t \in [0,T]$, $Z_t = \int_0^t e^{-\alpha(t-s)} dB_s$ is distributed as per $Z_t \sim N\left(0, \frac{1-e^{-2\alpha t}}{2\alpha} I_d\right)$. Note $\frac{1-e^{-2\alpha t}}{2\alpha} \leq \frac{1}{2\alpha}$. Thus by concentration of the norm of a $d$-dimensional Gaussian (see e.g. Theorem 3.1.1, Vershynin (2018)), we have for $\alpha \geq \alpha_{A.6}$ for a large enough universal constant $\alpha_{A.6}$, that $\mathbb{P}\left(\|Z_t\| > 0.03\sqrt{d}\right) \leq e^{-d}$. Consequently a Union Bound and our condition on $T$ gives, taking $c_{A.6} \leq \frac{1}{2}$,

$$\mathbb{P}(E) \leq \sum_{t=0}^{T\alpha} \mathbb{P}\left(\|Z_{t/\alpha}\| > 0.03\sqrt{d}\right) \leq (T\alpha + 1)e^{-d} \leq 2\alpha e^{-2c_{A.6}d}. \tag{15}$$

Next we upper bound the probability of the event $E^C \cap \bigcup_{t=0}^{T\alpha-1} \left\{ \sup_{h \in [0, \frac{1}{\alpha}]} \|Z_{\frac{t}{\alpha}+h} - Z_{\frac{t}{\alpha}}\| > 0.07\sqrt{d} \right\}$. Fixing $t \in \{0,1,\ldots,T\alpha-1\}$ in what follows, we write

$$Z_{\frac{t}{\alpha}+h} - Z_{\frac{t}{\alpha}} = \int_0^{\frac{t}{\alpha}} e^{-\alpha(\frac{t}{\alpha}-s)} \left(e^{-\alpha h} - 1\right) dB_s + \int_{\frac{t}{\alpha}}^{\frac{t}{\alpha}+h} e^{-\alpha(\frac{t}{\alpha}+h-s)} dB_s$$

$$= \left(e^{-\alpha h} - 1\right) Z_{\frac{t}{\alpha}} + \int_{\frac{t}{\alpha}}^{\frac{t}{\alpha}+h} e^{-\alpha(\frac{t}{\alpha}+h-s)} dB_s.$$

Note under $E^C$, we have $\|Z_{\frac{t}{\alpha}}\| \leq 0.03\sqrt{d}$. Since $\left| e^{-\alpha h} - 1 \right| \leq 1$, it follows that

$$\left\{ \sup_{h \in [0, \frac{1}{\alpha}]} \|Z_{\frac{t}{\alpha}+h} - Z_{\frac{t}{\alpha}}\| > 0.07\sqrt{d} \right\} \implies \left\{ \sup_{h \in [0, \frac{1}{\alpha}]} \left\| \int_{\frac{t}{\alpha}}^{\frac{t}{\alpha}+h} e^{-\alpha(\frac{t}{\alpha}+h-s)} dB_s \right\| > 0.04\sqrt{d} \right\}.$$

Let $u = \alpha(s - \frac{t}{\alpha})$ and $\tilde{B}_u := \sqrt{\alpha}\left(B_{\frac{t}{\alpha}+\frac{u}{\alpha}} - B_{\frac{t}{\alpha}}\right)$ for $u \geq 0$. Thus $\tilde{B}_u$ is a standard Brownian motion and $d\tilde{B}_u = \sqrt{\alpha} \cdot dB_{\frac{t}{\alpha}+\frac{u}{\alpha}}$. Now for any $h \in [0, \frac{1}{\alpha}]$, we may write

$$\int_{\frac{t}{\alpha}}^{\frac{t}{\alpha}+h} e^{-\alpha(\frac{t}{\alpha}+h-s)} dB_s = \int_0^{\alpha h} e^{-\alpha h + u} dB_{\frac{t}{\alpha}+\frac{u}{\alpha}} = \frac{e^{-\alpha h}}{\sqrt{\alpha}} \int_0^{\alpha h} e^u d\tilde{B}_u.$$

We apply Lemma A.7, and take $\alpha_{A.6} \geq \frac{1}{0.04^2 K_{A.7}^2}$ and $c_{A.6} \leq \frac{1}{2} c_{A.7}$. Thus as $[0, \alpha h] \subseteq [0,1]$,

$$\mathbb{P}\left(\sup_{h \in [0, \frac{1}{\alpha}]} \left\| \int_{\frac{t}{\alpha}}^{\frac{t}{\alpha}+h} e^{-\alpha(\frac{t}{\alpha}+h-s)} dB_s \right\| > 0.04\sqrt{d}\right) \leq \mathbb{P}\left(\sup_{h \in [0, \frac{1}{\alpha}]} \left\| \int_0^{\alpha h} e^u d\tilde{B}_u \right\| > 0.04\sqrt{\alpha d}\right)$$

$$\leq \mathbb{P}\left(\sup_{t' \in [0,1]} \left\| \int_0^{t'} e^u d\tilde{B}_u \right\| > 0.04\sqrt{\alpha d}\right)$$

$$\leq e^{-c_{A.7}d} \leq e^{-2c_{A.6}d}.$$

A Union Bound thus gives

$$\mathbb{P}\left(E^C \cap \bigcup_{t=0}^{T\alpha-1} \left\{ \sup_{h \in [0,\frac{1}{\alpha}]} \|Z_{\frac{t}{\alpha}+h} - Z_{\frac{t}{\alpha}}\| > 0.07\sqrt{d} \right\}\right) \le T\alpha \cdot e^{-2c_{A.6}d} \le \alpha e^{-1.5c_{A.6}d}. \tag{16}$$

Combining (14), (15), (16) proves Lemma A.6.

## B. Proofs of Theorems 3.1 and 3.7

### B.1. Proof of Theorem 3.1

We prove each of the claimed assertions in Theorem 3.1 as follows.

**Far from $\pi_{\mathrm{tar}}$:** Let $X_t(x_0)$ denote the law of (5) when initialized at $x_0$. Recall the crux is to prove Lemma 3.4, which we do using Lemma A.6.

*Proof of Lemma 3.4.* For all $0 \le t \le \tau(x_0)$, $\hat{s}(X_t(x_0)) = -\alpha X_t(x_0)$. Thus we remark that the SDE (5) and the SDE (8) have the same law through time $\tau(x_0)$. Let $\bar{X}_t(x_0)$ denote the law of $\bar{X}_t$ from (8) when initialized at $x_0$. Let $\bar{\tau}(x_0) := \inf_{t \ge 0} \{\|\bar{X}_t(x_0)\| \ge 4\sqrt{d}\}$. By the above, $\bar{\tau}(x_0) \equiv \tau(x_0)$ as random variables. Solving the SDE (8) explicitly gives

$$\bar{X}_t = e^{-\alpha t}x_0 + \sqrt{2} \int_0^t e^{-\alpha(t-s)}\mathrm{d}B_s.$$

If $\tau(x_0) \le T$, as argued above we must have $\bar{\tau}(x_0) \le T$. Therefore there exists a $t \in [0, T]$ such that

$$4\sqrt{d} \le \|\bar{X}_t(x_0)\| \le \|e^{-\alpha t}x_0\| + \sqrt{2}\left\|\int_0^t e^{-\alpha(t-s)}\mathrm{d}B_s\right\|$$

$$\le 1.1\sqrt{d} + \sqrt{2}\left\|\int_0^t e^{-\alpha(t-s)}\mathrm{d}B_s\right\|,$$

where we use that $\|x_0\| \le 1.1\sqrt{d}$. This means that $\left\|\int_0^t e^{-\alpha(t-s)}\mathrm{d}B_s\right\| > 2\sqrt{d}$, thus

$$\{\tau(x_0) \le T\} \subseteq \left\{ \sup_{t \in [0,T]} \left\|\int_0^t e^{-\alpha(t-s)}\mathrm{d}B_s\right\| > 2\sqrt{d} \right\}.$$

By Lemma A.6, the latter event has probability at most $3\alpha e^{-c_{A.6}d}$, proving Lemma 3.4. $\square$

Now consider any $T \le e^{c_{A.6}d/2}$. Let $\hat{\pi}_1$, $\hat{\pi}_2$ denote the law of (5) after time $T$ conditioned on $\|X_0\| \le 1.1\sqrt{d}$, $\|X_0\| > 1.1\sqrt{d}$ respectively. Let $p = \mathbb{P}_{X_0 \sim N(0,I_d)}(\|X_0\| \le 1.1\sqrt{d})$ and let $A := \{x : \|x\| \ge 4\sqrt{d}\}$. By Lemma A.1 on Gaussian concentration of the norm, $p \ge 1 - e^{-\Omega(d)}$ and $\pi_{\mathrm{tar}}(A) \ge 1 - e^{-\Omega(d)}$. Thus

$$\mathsf{TV}\big(\mathcal{L}(X_T), \pi_{\mathrm{tar}}\big)$$
$$= \sup_{E \subseteq \mathbb{R}^d} \left| p\hat{\pi}_1(E) + (1-p)\hat{\pi}_2(E) - \pi_{\mathrm{tar}}(E) \right|$$
$$\ge \left| p\hat{\pi}_1(A) + (1-p)\hat{\pi}_2(A) - \pi_{\mathrm{tar}}(A) \right|$$
$$\ge \left| p\big|\hat{\pi}_1(A) - \pi_{\mathrm{tar}}(A)\big| - (1-p)\big|\hat{\pi}_2(A) - \pi_{\mathrm{tar}}(A)\big| \right|$$
$$\ge (1 - e^{-\Omega(d)})\big(1 - e^{-\Omega(d)} - \hat{\pi}_1(A)\big) - e^{-\Omega(d)},$$

where we used Triangle Inequality in the second inequality above. By Lemma 3.4, since $X_T(x_0) \in A$ implies that $\tau(x_0) \le T$, we have $\hat{\pi}_1(A) \le 3\alpha e^{-c_{A.6}d/2}$. Plugging into the above, and as $c_{A.6}$ is a universal constant, we have for $d \ge d_0$ that

$$\mathsf{TV}\big(\mathcal{L}(X_T), \pi_{\mathrm{tar}}\big) \ge 1 - e^{-\Omega(d)}.$$

**Small $L^p$ error:** Note $\hat{s}(x) - \nabla \log \pi_{\mathrm{tar}} = 0$ if $\|x\| \geq 5\sqrt{d}$. Otherwise if $\|x\| < 5\sqrt{d}$, as $\psi(x) \in [0, 1]$, $\|\hat{s}(x) - \nabla \log \pi_{\mathrm{tar}}\| \leq \alpha\|x\| < 5\alpha\sqrt{d}$. Note $\pi_{\mathrm{tar}}\left(\{\|x\| < 5\sqrt{d}\}\right) \leq \pi_{\mathrm{tar}}\left(\{\|x - \mu\| \geq 2\sqrt{d}\}\right) \leq e^{-\Omega(d)}$ by Gaussian concentration of the norm (Lemma A.1). Thus as $\alpha = \alpha_{A.6}$ is a universal constant, for $d \geq d_0(p)$,

$$\mathop{\mathbb{E}}_{x \sim \pi_{\mathrm{tar}}} \left[\|\hat{s}(x) - \nabla \log \pi_{\mathrm{tar}}(x)\|^p\right]^{1/p} \leq \left((5\alpha)^p d^{p/2} e^{-\Omega(d)}\right)^{1/p} \leq e^{-\Omega(d)}.$$

**Lipschitz $\hat{s}$:** It is easy to check $\hat{s}(x)$ is Lipschitz whenever $\|x\| < 4\sqrt{d}$ or $\|x\| > 5\sqrt{d}$. In Lemma A.5, we construct $\psi(\cdot) \in [0, 1]$ to be $H$-Lipschitz for a universal constant $H > 0$, and $\psi(\cdot)$ is infinitely differentiable. For $x$ such that $4\sqrt{d} \leq \|x\| \leq 5\sqrt{d}$, we can compute

$$\nabla \hat{s}(x) = -\alpha\psi\left(\frac{x}{\sqrt{d}}\right) - \frac{\alpha x}{\sqrt{d}}\nabla\psi\left(\frac{x}{\sqrt{d}}\right) - \left(1 - \psi\left(\frac{x}{\sqrt{d}}\right)\right) + \frac{x}{\sqrt{d}}\nabla\psi\left(\frac{x}{\sqrt{d}}\right).$$

Thus as $\alpha = \alpha_{A.6}$ is a universal constant, $\hat{s}$ is Lipschitz with a universal constant bound on its Lipschitz constant.

This concludes the proof of Theorem 3.1.

### B.2. Proof of Theorem 3.7

Here we prove Theorem 3.7. We first prove a useful helper result.

**Lemma B.1.** *Suppose $n = poly(d)$ and $x_1, \ldots, x_n$ are in general position. Let $A := \{x : \|x - x_i\| \geq 0.16\sqrt{d} \text{ for all } 1 \leq i \leq n\}$. Then $\pi_{\mathrm{tar}}(A) \geq 1 - e^{-\Omega(d)}$ for $d \geq d_0$, where $d_0$ is a universal constant.*

*Proof.* Equivalently, we will upper bound $\pi_{\mathrm{tar}}(A^C) \leq e^{-\Omega(d)}$. Note $A^C = \{x : \|x - x_i\| \leq 0.16\sqrt{d} \text{ for some } x_i\}$. For any given $i$, since the $x_1, \ldots, x_n$ are in general position we have $2\sqrt{d} \geq \|x_i\| \geq 0.5\sqrt{d}$, and a direct geometric argument gives

$$\left\{x : \|x - x_i\| \leq 0.16\sqrt{d}\right\} \subseteq \left\{x : \left|\frac{\langle x_i, x\rangle}{\|x_i\|\|x\|}\right| \leq \frac{0.15}{0.5}, \|x\| \leq 2.34\sqrt{d}\right\}$$

$$\subseteq \left\{x : \left|\frac{\langle x_i, x\rangle}{\|x_i\|}\right| \leq 0.71\sqrt{d}\right\}.$$

Note $f(x) = \frac{\langle x_i, x\rangle}{\|x_i\|}$ is a 1-Lipschitz function of $x$. Also by spherical symmetry of the standard Gaussian $\pi_{\mathrm{tar}}$, we have $\mathbb{E}_{X \sim \pi_{\mathrm{tar}}}\left[\frac{\langle x_i, X\rangle}{\|x_i\|}\right] = 0$. Thus Lemma A.2 gives

$$\pi_{\mathrm{tar}}\left(\{x : \|x - x_i\| \leq 0.16\sqrt{d}\}\right) = \mathbb{P}_{X \sim N(0, I_d)}\left(\left|\frac{\langle x_i, X\rangle}{\|x_i\|}\right| \leq 0.71\sqrt{d}\right) \leq 2e^{-\Omega(d)}.$$

Consequently, a Union Bound gives

$$\pi_{\mathrm{tar}}(A^C) \leq 2ne^{-\Omega(d)} \leq \mathrm{poly}(d)e^{-\Omega(d)} \leq e^{-\Omega(d)}$$

for $d \geq d_0$ larger than a universal constant. This proves the Lemma. $\qquad\square$

Now, we prove each of the claimed assertions in Theorem 3.7 as follows.

**Far from $\pi_{\mathrm{tar}}$ if the $x_1, \ldots, x_n$ are in general position:** Let $X_t(x_0)$ denote the law of

$$\mathrm{d}X_t = \hat{s}(X_t)\,\mathrm{d}t + \sqrt{2}\,\mathrm{d}B_t, \qquad X_0 = x_0. \tag{17}$$

The crux is to show the following Lemma.

**Lemma B.2.** *Suppose the $x_1, \ldots, x_n$ are in general position. Suppose $d \geq K_{A.6}$ where $K_{A.6} > 0$ is a universal constant and consider any $T \leq e^{c_{A.6}d/2}$. Let $\tau(x_0) = \inf_{t \geq 0}\left\{\|X_t(x_0) - x_0\| \geq 0.15\sqrt{d}\right\}$. Then for any $1 \leq i \leq n$, we have with probability at least $1 - 3\alpha e^{-c_{A.6}d}$ that $\tau(x_i) > T$.*

*Proof of Lemma B.2.* For all $0 \leq t \leq \tau(x_i)$, $\hat{s}(x) = -\alpha(x - x_i)$. Thus we remark that the SDE (17) with initialization $x_0 = x_i$ and the following SDE (18) have the same law through time $\tau(x_i)$:

$$d\bar{X}_t = -\alpha(\bar{X}_t - x_i)dt + \sqrt{2}dB(t) \text{ where } \bar{X}(0) = x_i. \tag{18}$$

Let $\bar{X}_t(x_0)$ denote the law of $\bar{X}_t$ when initialized at $x_0$. Let $\bar{\tau}(x_0) := \inf_{t \geq 0}\left\{\|\bar{X}_t(x_0) - x_0\| \geq 0.15\sqrt{d}\right\}$. By the above remark we have $\bar{\tau}(x_i) \equiv \tau(x_i)$ as random variables. Moreover, (18) has the same law as the fixed translation $x_i$ plus a random variable drawn from the law of

$$d\tilde{X}_t = -\alpha\tilde{X}_t dt + \sqrt{2}dB(t) \text{ where } \tilde{X}(0) = 0. \tag{19}$$

Let $\tilde{X}_t(x_0)$ denote the law of $\tilde{X}_t$ when initialized at $x_0$. Hence $\bar{\tau}(x_i), \tau(x_i), \tilde{\tau}(0)$ have the same law where $\tilde{\tau}(x_0) := \inf_{t \geq 0}\left\{\|\tilde{X}_t(x_0)\| \geq 0.15\sqrt{d}\right\}$. Solving the SDE (19) explicitly we obtain

$$\tilde{X}_t = \sqrt{2}\int_0^t e^{-\alpha(t-s)}dB_s.$$

If $\tau(x_0) \leq T$, as argued above we must have $\tilde{\tau}(x_0) \leq T$. Therefore there exists a $t \in [0, T]$ such that

$$\left\|\int_0^t e^{-\alpha(t-s)}dB_s\right\| \geq \frac{0.15}{\sqrt{2}}\sqrt{d} > 0.1\sqrt{d}.$$

By Lemma A.6, this latter event has probability at most $3\alpha e^{-c_{A.6}d}$, proving Lemma B.2. $\qquad\square$

Now consider any $T \leq e^{c_{A.6}d/2}$. Let $\hat{\pi}_i$ denote the law of (17) after time $T$ with initialization $x_i$. Letting $\hat{\pi}$ denote the law of (9) after time $T$, which is initialized at $\frac{1}{n}\sum_{i=1}^n \delta_{x_i}$, we note for all $E \subseteq \mathbb{R}^d$ that

$$\hat{\pi}(E) = \sum_{i=1}^n \frac{1}{n}\hat{\pi}_i(E).$$

Let $A := \left\{x : \|x - x_i\| \geq 0.16\sqrt{d} \text{ for all } 1 \leq i \leq n\right\}$. Thus

$$\mathsf{TV}\big(\mathcal{L}(X_T), \pi_{\text{tar}}\big) = \sup_{E \subseteq \mathbb{R}^d}\left|\sum_{i=1}^n \frac{1}{n}\hat{\pi}_i(E) - \pi_{\text{tar}}(E)\right|$$

$$\geq \left|\sum_{i=1}^n \frac{1}{n}\hat{\pi}_i(A) - \pi_{\text{tar}}(A)\right|.$$

Consider any $1 \leq i \leq n$. By Lemma B.2, since $X_t(x_i) \in A$ implies that $\tau(x_i) \leq T$, we have $\hat{\pi}_i(A) \leq 3\alpha e^{-c_{A.6}d/2}$. By Lemma B.1, $\pi_{\text{tar}}(A) \geq 1 - e^{-\Omega(d)}$. Plugging into the above, and as $c_{A.6}$ is a universal constant, we have for $d \geq d_0$ that

$$\mathsf{TV}\big(\mathcal{L}(X_T), \pi_{\text{tar}}\big) \geq \left|\sum_{i=1}^n \frac{1}{n}\hat{\pi}_i(A) - \pi_{\text{tar}}(A)\right|$$

$$\geq 1 - e^{-\Omega(d)} - n \cdot \frac{1}{n} \cdot 3\alpha e^{-c_{A.6}d/2}$$

$$\geq 1 - e^{-\Omega(d)},$$

as desired.

**Small $L^p$ error if $x_1, \ldots, x_n$ are in general position:** Let $A := \left\{x : \|x - x_i\| \geq 0.16\sqrt{d} \text{ for all } 1 \leq i \leq n\right\}$. Note $\hat{s}(x) - \nabla\log\pi_{\text{tar}} = 0$ if $x \in A$. Otherwise if $x \in A^C$, as $\psi(x) \in [0, 1]$, $\|\hat{s}(x) - \nabla\log\pi_{\text{tar}}\| \leq \alpha\|x - x_i\| < 3\alpha\sqrt{d}$ as $\|x_i\| \leq 2\sqrt{d}$ by definition of general position. By Lemma B.1, $\pi_{\text{tar}}(A^C) \leq e^{-\Omega(d)}$. Thus as $\alpha = \alpha_{A.6}$ is a universal constant, for $d \geq d_0(p)$,

$$\mathop{\mathbb{E}}_{x \sim \pi_{\text{tar}}}\left[\|\hat{s}(x) - \nabla\log\pi_{\text{tar}}(x)\|^p\right]^{1/p} \leq \left((3\alpha)^p d^{p/2} e^{-\Omega(d)}\right)^{1/p} \leq e^{-\Omega(d)}.$$

**Lipschitz $\hat{s}$ if $x_1, \ldots, x_n$ are in general position:** Again let $A := \{x : \|x - x_i\| \geq 0.16\sqrt{d} \text{ for all } 1 \leq i \leq n\}$. It is easy to check $\hat{s}(x)$ is Lipschitz whenever $x \in A$ or $\|x - x_i\| \leq 0.15\sqrt{d}$ for some $i$ (which must be unique if $x_1, \ldots, x_n$ are in general position). In Lemma A.5, we construct $\psi(\cdot) \in [0, 1]$ to be $H$-Lipschitz for a universal constant $H > 0$, and $\psi(\cdot)$ is infinitely differentiable. For $x$ such that $0.15\sqrt{d} \leq \|x - x_i\| \leq 0.16\sqrt{d}$ for some $i$ (which must be unique if $x_1, \ldots, x_n$ are in general position), we can compute

$$\nabla \hat{s}(x) = -\alpha \psi\Big(\frac{100(x - x_i) - 11}{\sqrt{d}}\Big) - \frac{100\alpha x}{\sqrt{d}} \nabla \psi\Big(\frac{100(x - x_i) - 11}{\sqrt{d}}\Big)$$
$$- \Big(1 - \psi\Big(\frac{100(x - x_i) - 11}{\sqrt{d}}\Big)\Big) + \frac{100x}{\sqrt{d}} \nabla \psi\Big(\frac{100(x - x_i) - 11}{\sqrt{d}}\Big).$$

Thus as $\alpha = \alpha_{A.6}$ is a universal constant, $\hat{s}$ is Lipschitz with a universal constant bound on its Lipschitz constant.

$x_1, \ldots, x_n$ **are in general position with high probability:** Since each $x_i \sim N(0, I_d)$, $x_i - x_j \sim N(0, 2I_d)$. By Lemma A.1, we have $\|x_i - x_j\| \geq 0.4\sqrt{d}$ with probability at least $1 - e^{-\Omega(d)}$. Another direct application of Lemma A.1 shows that each $x_i$ is such that $0.5\sqrt{d} \leq \|x_i\| \leq 2\sqrt{d}$ with probability at least $1 - e^{-\Omega(d)}$. Thus as $n = \text{poly}(d)$, a Union Bound gives that $x_1, \ldots, x_n$ are in general position with probability at least $1 - \big(\binom{n}{2} + n\big)e^{-\Omega(d)} \geq 1 - e^{-\Omega(d)}$ for $d \geq d_0$, where $d_0$ is a universal constant.

This concludes the proof of Theorem 3.7.

## C. Proofs for Subsection 3.3

Here we prove Theorem 3.11. Throughout this proof, we let $1\{\cdot\}$ denote the indicator function, and define the convex cone $\text{Cone}_{\theta, u}$ as follows (Boyd & Vandenberghe, 2004): for a half–angle $\theta \in (0, \pi/2)$ and unit vector $u \in \mathbb{R}^d$,

$$\text{Cone}_{\theta, u} := \Big\{y \in \mathbb{R}^d \setminus \{0\} : \big\langle \frac{y}{\|y\|}, u \big\rangle \geq \cos\theta\Big\} \tag{20}$$
$$= \{y : g(y) \geq 0\}, \quad g(y) := \langle y, u \rangle - \|y\| \cos\theta.$$

**Constructing $\theta, u$:** First, for $k \in \mathbb{N}_+$, define $s_k(x) = \nabla \log \pi_{\text{tar}}(x) 1\{\|x\| > k\}$. Note that $s_k(x) \to 0$ as $k \to \infty$ and $\|s_k(x)\|^2 \leq \|\nabla \log \pi_{\text{tar}}(x)\|^2 \in L^2(\pi_{\text{tar}})$. By Lebesgue's dominated convergence theorem, we have $\mathbb{E}_{\pi_{\text{tar}}}[\|s_k(x)\|^2] \to 0$ as $k \to \infty$. Similarly, we have $\mathbb{P}_{\pi_{\text{tar}}}(\{x : \|x\| > k\}) \to 0$ as $k \to \infty$. We then choose a sufficiently large $k_0$, such that

$$\mathbb{E}_{\pi_{\text{tar}}}[\|s_{k_0}(x)\|^2] \leq \varepsilon_{\text{score}}^2/8, \qquad \mathbb{P}_{\pi_{\text{tar}}}(\{x : \|x\| > k_0\}) \leq \min\{\varepsilon_{\text{score}}^2/10, \ \varepsilon_{\text{TV}}/2, \ 1/2\}.$$

Note that $\int 1\{\|x\| \leq k_0\}\|\nabla \log \pi_{\text{tar}}(x)\|^2 \pi_{\text{tar}}(x)\mathrm{d}x < \infty$ and $\int 1\{\|x\| \leq k_0\} \pi_{\text{tar}}(x)\mathrm{d}x < \infty$. By absolute continuity of the Lebesgue integral, there exists $\delta > 0$, such that for any set $A$ with Lebesgue measure no larger than $\delta$, we have $\int_A 1\{\|x\| \leq k_0\}\|\nabla \log \pi_{\text{tar}}(x)\|^2 \pi_{\text{tar}}(x)\mathrm{d}x \leq \varepsilon_{\text{score}}^2/8$ and $\int_A 1\{\|x\| \leq k_0\} \pi_{\text{tar}}(x)\mathrm{d}x \leq \min\{\varepsilon_{\text{score}}^2/8, \varepsilon_{\text{TV}}/2\}$.

We then choose $\theta \in (0, 2\pi)$ small enough, such that the Lebesgue measure of $\text{Cone}_{\theta, u} \cap \{x : \|x\| \leq k_0\}$ is no larger than $\delta$ for any unit vector $u \in \mathbb{R}^d$. Since $\mathbb{P}_{\pi_{\text{tar}}}(\{x : \|x\| \leq k_0\}) \geq 1/2$, then there exists a unit vector $u \in \mathbb{R}^d$, such that $\mathbb{P}_{\pi_{\text{tar}}}(\{x : \|x\| \leq k_0\} \cap \text{Cone}_{\theta, u}) > 0$.

**Constructing $\hat{s}$:** Let $\text{Cone}_{\theta, u}^\circ$ denote the open set that is $\text{Cone}_{\theta, u}$ minus its boundary. Let $K \subseteq \text{Cone}_{\theta, u}^\circ$ be closed ball of positive Lebesgue measure. We can clearly take a smaller closed ball $K' \subset K$ containing an open ball inside it, such that the containment within $K$ is strict. Let $B = \bigcap_{x \in K'}(K - x)$, where $K - x$ refers to translation. Note $B$ is nonempty, has positive Lebesgue measure, and contains an open ball. Hence if $x \in K'$ and $y - x \in B$, we have $y \in K$. We now define

$$\hat{s}(x) := \begin{cases} u & : x \in K' \\ \nabla \log \pi_{\text{tar}} & : x \notin K'. \end{cases}$$

Recall the definition of the SDE for $(X_t)_{t \geq 0}$ in terms of $\hat{s}$:

$$\mathrm{d}X_t = \hat{s}(X_t)\,\mathrm{d}t + \sqrt{2}\,\mathrm{d}B_t, \qquad X_0 \sim \pi_0.$$

Note the SDE given above has a unique strong solution by Zvonkin's Theorem (Zvonkin, 1974). Also note that $(X_t)_{t\geq 0}$ is a Markov process associated with natural filtration $(\mathcal{F}_t)_{t\geq 0}$, and moreover satisfies the strong Markov property since the SDE defining $X_t$ has a unique strong solution.

**Small $L^2$ error of $\hat{s}$:** With such $k_0, \theta, u$, since $K' \subseteq \text{Cone}_{\theta,u}$, we have

$$\mathbb{E}_{\pi_{\text{tar}}}[\|\hat{s}(x) - \nabla \log \pi_{\text{tar}}(x)\|^2]$$

$$\leq \int_{\text{Cone}_{\theta,u}} \|u - \nabla \log \pi_{\text{tar}}(x)\|^2 \pi_{\text{tar}}(x)\mathrm{d}x$$

$$\leq \int_{\text{Cone}_{\theta,u} \cap \{x:\|x\|>k_0\}} \|u - \nabla \log \pi_{\text{tar}}(x)\|^2 \pi_{\text{tar}}(x)\mathrm{d}x + \int_{\text{Cone}_{\theta,u} \cap \{x:\|x\|\leq k_0\}} \|u - \nabla \log \pi_{\text{tar}}(x)\|^2 \pi_{\text{tar}}(x)\mathrm{d}x$$

$$\leq 2 \int_{\text{Cone}_{\theta,u} \cap \{x:\|x\|>k_0\}} (1 + \|\nabla \log \pi_{\text{tar}}(x)\|^2) \pi_{\text{tar}}(x)\mathrm{d}x + 2 \int_{\text{Cone}_{\theta,u} \cap \{x:\|x\|\leq k_0\}} (1 + \|\nabla \log \pi_{\text{tar}}(x)\|^2) \pi_{\text{tar}}(x)\mathrm{d}x$$

$$= 2\mathbb{P}_{\pi_{\text{tar}}}(\{x:\|x\|>k_0\}) + 2\mathbb{P}_{\pi_{\text{tar}}}(\{x:\|x\|\leq k_0\} \cap \text{Cone}_{\theta,u}) + 2\int_{\|x\|>k_0} \|\nabla \log \pi_{\text{tar}}(x)\|^2 \pi_{\text{tar}}(x)\mathrm{d}x$$

$$+ 2\int_{\text{Cone}_{\theta,u} \cap \{x:\|x\|\leq k_0\}} \|\nabla \log \pi_{\text{tar}}(x)\|^2 \pi_{\text{tar}}(x)\mathrm{d}x$$

$$\leq \varepsilon_{\text{score}}^2.$$

**Far from $\pi_{\text{tar}}$ in Total Variation as $t \to \infty$:** We will prove this by showing that $\mathbb{P}_{\pi_{\text{tar}}}(\text{Cone}_{\theta,u}) \leq \varepsilon_{\text{TV}}$ and $\mathbb{P}_{\pi_t}(\text{Cone}_{\theta,u}) \to 1$ as $t \to \infty$. Together, these facts directly imply that $\liminf_{t\to\infty} \text{TV}(\pi_{\text{tar}}, \pi_t) \geq 1 - \varepsilon_{\text{TV}}$.

**Small $\mathbb{P}_{\pi_{\text{tar}}}(\text{Cone}_{\theta,u})$:** Since the Lebesgue measure of $\text{Cone}_{\theta,u} \cap \{x:\|x\| \leq k_0\}$ is no larger than $\delta$, we have

$$\mathbb{P}_{\pi_{\text{tar}}}(\text{Cone}_{\theta,u}) \leq \mathbb{P}_{\pi_{\text{tar}}}(\{x:\|x\|>k_0\}) + \mathbb{P}_{\pi_{\text{tar}}}(\text{Cone}_{\theta,u} \cap \{x:\|x\|\leq k_0\})$$

$$\leq \min\{\varepsilon_{\text{score}}^2/10, \varepsilon_{\text{TV}}/2, 1/2\} + \min\{\varepsilon_{\text{score}}^2/8, \varepsilon_{\text{TV}}/2\}$$

$$\leq \varepsilon_{\text{TV}}.$$

**Showing $\mathbb{P}_t(\text{Cone}_{\theta,u}) \to 1$ as $t \to \infty$:** Our proof builds on existing analyses of the exit time of Brownian motion with drift from a cone (Garbit & Raschel, 2014).

**Lemma C.1** (Garbit & Raschel (2014)). *Consider Brownian motion with a drift:*

$$\mathrm{d}X_t = u\,\mathrm{d}t + \sqrt{2}\,\mathrm{d}B_t, \qquad X_0 = x \in \text{Cone}_{\theta,u}^{\circ}.$$

*where $(B_t)_{t\geq 0}$ is a $d$-dimensional standard Brownian motion. Define $T = \inf_{t\geq 0}\{X_t \notin \text{Cone}_{\theta,u}\}$. Then $\mathbb{P}(T = \infty) = p_{\text{no exit}}(x) > 0$. Here, $p_{\text{no exit}} : \text{Cone}_{\theta,u}^{\circ} \mapsto \mathbb{R}_{>0}$ is continuous.*

Now, for any $x \in \mathbb{R}^d$, define the auxiliary process $(Y_t(x))_{t\geq 0}$ by the following SDE that is a Brownian motion with drift:

$$\mathrm{d}Y_t(x) = u\,\mathrm{d}t + \sqrt{2}\,\mathrm{d}B_t, \qquad Y_0 = x. \tag{21}$$

We define

$$\alpha_{K'} := \inf_{x\in K'} \mathbb{E}\left[1\{Y_1(x) - x \in B\}1\{(Y_t)_{t\in[0,1]} \in K'\}\right].$$

We claim $\alpha_{K'} > 0$. Letting $B'$ be an open ball strictly contained in $B$ and letting $K''$ be an open ball strictly contained in $K'$, we have $\alpha_{K'} \geq \inf_{x\in K'} \mathbb{E}\left[1\{Y_1(x) - x \in B'\}1\{(Y_t)_{t\in[0,1]} \in K''\}\right]$. Next by classical results on Brownian motion with drift, the map $x \to \mathbb{E}\left[1\{Y_1(x) - x \in B'\}1\{(Y_t)_{t\in[0,1]} \in K''\}\right]$ is continuous in $x$ and strictly positive as $B'$ and $K''$ are open. In particular as $K'$ is compact, we obtain strict positivity of this map over $K'$ by the Stroock-Varadhan Support Theorem (Stroock & Varadhan, 1972), and we obtain continuity as a Brownian motion with drift satisfies the Strong Feller property (see e.g. 24.1, Rogers & Williams (2000)). Thus $\alpha_{K'} > 0$.

We inductively define the following stopping times:

$$T_1 = \inf\{t \geq 0 : X_t \in K'\}, \qquad \tau_1 = \inf\{t \geq T_1 + 1 : X_t \notin \text{Cone}_{\theta,u}\},$$
$$T_k = \inf\{t \geq \tau_{k-1} : X_t \in K'\}, \qquad \tau_k = \inf\{t \geq T_k + 1 : X_t \notin \text{Cone}_{\theta,u}\}, \qquad \text{for } k \geq 2.$$

We then show that $\mathbb{P}(T_1 < \infty) = 1$. To prove this result, we introduce an auxiliary process $(Y_t)_{t \geq 0}$ defined by the SDE below:

$$\mathrm{d}Y_t = \nabla \log \pi_{\mathrm{tar}}(Y_t)\mathrm{d}t + \sqrt{2}\mathrm{d}B_t, \qquad Y_0 = X_0.$$

Define $T_1^Y = \inf\{t \geq 0 : Y_t \in K'\}$. Since the processes $(X_t)_{t \geq 0}$ and $(Y_t)_{t \geq 0}$ coincide up to hitting $K'$, it follows that $T_1 = T_1^Y$, and $\mathbb{P}(T_1 < \infty) = \mathbb{P}(T_1^Y < \infty)$. Recall $Y_t \overset{d}{\to} \pi_{\mathrm{tar}}$. Combined with Assumption 3.10, the associated diffusion is non-explosive, irreducible, and positive Harris recurrent with invariant distribution $\pi_{\mathrm{tar}}$. Since $\pi_{\mathrm{tar}}(K') > 0$ as $K'$ has positive Lebesgue measure, it follows that $\mathbb{P}(T_1^Y < \infty) = 1$. Hence $\mathbb{P}(T_1 < \infty) = 1$.

We now look at the probability $\mathbb{P}(\tau_1 = \infty | T_1 < \infty)$. Note that because $X_{T_1} \in K'$, if $X_{T_1+1} - X_{T_1} \in B$, we have $X_{T_1+1} \in K$. Applying Lemma C.1 and the compactness of $K$, we have $\inf_{x \in K} p_{\mathrm{no\,exit}}(x) > 0$. Thus by the strong Markov property,

$$
\begin{aligned}
\mathbb{P}(\tau_1 = \infty) &\geq \mathbb{E}\left[1\{X_{T_1+1} \in K\}p_{\mathrm{no\,exit}}(X_{T_1+1})\right] \\
&\geq \mathbb{E}\left[1\{X_{T_1+1} - X_{T_1} \in B\}1\{(X_t)_{t \in [T_1, T_1+1]} \in K'\}p_{\mathrm{no\,exit}}(X_{T_1+1})\right] \\
&\geq \mathbb{E}\left[1\{X_{T_1+1} - X_{T_1} \in B\}1\{(X_t)_{t \in [T_1, T_1+1]} \in K'\}\inf_{x \in K} p_{\mathrm{no\,exit}}(x)\right] \\
&= \inf_{x \in K} p_{\mathrm{no\,exit}}(x) \cdot \mathbb{E}\left[1\{X_{T_1+1} - X_{T_1} \in B\}1\{(X_t)_{t \in [T_1, T_1+1]} \in K'\}\right].
\end{aligned}
$$

Note that within $K'$, we have

$$\mathrm{d}X_t = u\mathrm{d}t + \sqrt{2}\mathrm{d}B_t.$$

Consequently since $X_{T_1} \in K'$, we have by the strong Markov property that

$$
\begin{aligned}
\mathbb{E}&\left[1\{X_{T_1+1} - X_{T_1} \in B\}1\{(X_t)_{t \in [T_1, T_1+1]} \in K'\}\right] \\
&= \mathbb{E}\left[\mathbb{E}_{X_{T_1}}\left[1\{X_{T_1+1} - X_{T_1} \in B\}1\{(X_t)_{t \in [T_1, T_1+1]} \in K'\}\right]\right] \\
&\geq \inf_{x \in K'} \mathbb{E}\left[1\{Y_1(x) - x \in B\}1\{(Y_t)_{t \in [0,1]} \in K'\}\right] \\
&= \alpha_{K'} > 0,
\end{aligned}
$$

where $\mathbb{E}_{X_{T_1}}[\cdot]$ denotes expectation w.r.t. $Y_t(X_{T_1})$, the process above started at $X_{T_1}$, where $Y_t(x)$ was defined in (21). Here we use that $\alpha_{K'} > 0$ as justified earlier. Combining the steps above gives

$$\mathbb{P}(\tau_1 = \infty) \geq \alpha_{K'} \inf_{x \in K} p_{\mathrm{no\,exit}}(x) > 0.$$

Similarly, for every $k \in \mathbb{N}_+$ with $k \geq 2$, we can prove that

$$\mathbb{P}(T_k < \infty \mid \tau_{k-1} < \infty) = 1, \qquad \mathbb{P}(\tau_k = \infty \mid T_k < \infty) \geq \alpha_{K'} \inf_{x \in K} p_{\mathrm{no\,exit}}(x) > 0.$$

Combining the above, we obtain $\mathbb{P}(\tau_k = \infty) \geq 1 - (1 - \alpha_{K'} \inf_{x \in K} p_{\mathrm{no\,exit}}(x))^k \to 1$ as $k \to \infty$. Note that for any $k \in \mathbb{N}_+$ and any $t \geq 0$, we have

$$\mathbb{P}_{\pi_t}(\mathrm{Cone}_{\theta,u}) \geq \mathbb{P}(\tau_k = \infty, t \geq T_k + 1).$$

Therefore, $\liminf_{t \to \infty} \mathbb{P}_{\pi_t}(\mathrm{Cone}_{\theta,u}) \geq \liminf_{t \to \infty} \mathbb{P}(\tau_k = \infty, t \geq T_k+1) = \mathbb{P}(\tau_k = \infty)$, where the last equality follows by Monotone Convergence Theorem. This holds for any $k \in \mathbb{N}_+$, so combining the above, we have $\lim_{t \to \infty} \mathbb{P}_{\pi_t}(\mathrm{Cone}_{\theta,u}) = 1$. The proof is done.

## D. Simulation details

**Learning the score:** We follow the DDPM setup for learning an estimate of the score function (see Algorithm 1 of Ho et al. (2020)). In particular, we consider a linear noise schedule at 1000 noise levels, where for each $1 \leq t \leq 1000$, $\beta_t = \frac{1001-t}{1000} \cdot 10^{-4} + \frac{t-1}{1000} \cdot 10^{-2}$. We train a NN $\hat{s}_\theta(x,t)$ that takes as input $x \in \mathbb{R}^d$ and $t \in \{1, \ldots, 1000\}$ to learn the

added noise (the manner in which noise is added will be described next). The NN has three hidden layers, where the hidden layers are of dimension 256 for Gaussian $\pi_{\text{tar}}$ and are of dimension 512 for GMM $\pi_{\text{tar}}$.

We create a training set of 10000 samples by drawing 1000 i.i.d. samples from $\pi_{\text{tar}}$ and duplicating each sample 10 times. The NN is trained with the Adam (Kingma & Ba, 2015) optimizer run for 150,000 epochs with learning rate $10^{-3}$. The batch size is 10,000, the entire dataset. In training we perform the following procedure. We draw $t$ uniformly from $\{1, \ldots, 200\}$. Let $\bar{\alpha}_t = \prod_{t'=1}^t (1 - \beta_{t'})$. Then for all $x_i$ in the batch (which here is the whole dataset), we let $x_i' = \sqrt{\bar{\alpha}_t} x_i + \sqrt{1 - \bar{\alpha}_t} \varepsilon_i$ where $\varepsilon_i \sim N(0, I_d)$ are drawn i.i.d. for each $x_i$. We aim to minimize the mean square loss between $\hat{s}_\theta(x_i', t)$ and $\varepsilon_i$ over the batch (again, which here is the whole dataset). As we mentioned earlier, these design choices were made to encourage a $\hat{s}$ that overfits to the 1000 i.i.d. samples from $\pi_{\text{tar}}$. We focus on $t$ drawn uniformly from $\{1, \ldots, 200\}$ – which are lower noise levels – because we aim to learn an approximation to $\nabla \log \pi_{\text{tar}}$, which corresponds to no noise added to the training data.

**Sampling algorithms:** We run Langevin dynamics by implementing ULA as written in (2) with fixed step-size 0.0025 for 1000 iterations. As mentioned earlier, we use 1000 iterations to simulate a poly$(d)$ timescale (recall here $d = 25$ or $50$), and as score-based generative models rarely use more than 1000 denoiser evaluations in practice with fewer denoiser evaluations being considered desirable (Karras et al., 2022). To use our trained NN $\hat{s}_\theta(x, t)$ to approximate the score function $\nabla \log \pi_{\text{tar}}$, we approximate $\nabla \log \pi_{\text{tar}}(x)$ by $-\frac{\hat{s}_\theta(x, 10)}{\sqrt{1 - \bar{\alpha}_{10}}}$. We set $t = 10$ here, a low noise level, as $\nabla \log \pi_{\text{tar}}$ corresponds to no noise added. We did not set $t$ as low as possible and rather chose the noise level $t = 10$ because it was better represented in training, leading to a $\hat{s}_\theta(x, t)$ being a more faithful score estimate for such $t$.

**Compute:** Our experiments were all run on a single RTX A6000 GPU.

**Additional results:** We also plot the empirical Wasserstein distance between the produced samples and $\pi_{\text{tar}}$ for the case when $\pi_{\text{tar}}$ is a Gaussian. The plot is given in Figure 4. As with the plot for empirical Wasserstein distance between the produced samples and $\pi_{\text{tar}}$ for Gaussian $\pi_{\text{tar}}$, the Wasserstein distance is computed with the Sinkhorn loss (Cuturi, 2013). In particular, for both of these computations, we draw 5000 samples from $\pi_{\text{tar}}$. We then run 10 trials, and for each trial we subsample 500 points from the produced sample and the 5000 sampled points from $\pi_{\text{tar}}$, computing the Sinkhorn loss between these two sets of 500 points. The Sinkhorn loss is applied with $p = 2$ and blur scaling 0.01. Note we did not plot KL for GMM $\pi_{\text{tar}}$ since estimating KL from samples is unstable and expensive in high dimensions, and fitting the produced samples to a Gaussian does not make sense when $\pi_{\text{tar}}$ is not a Gaussian.

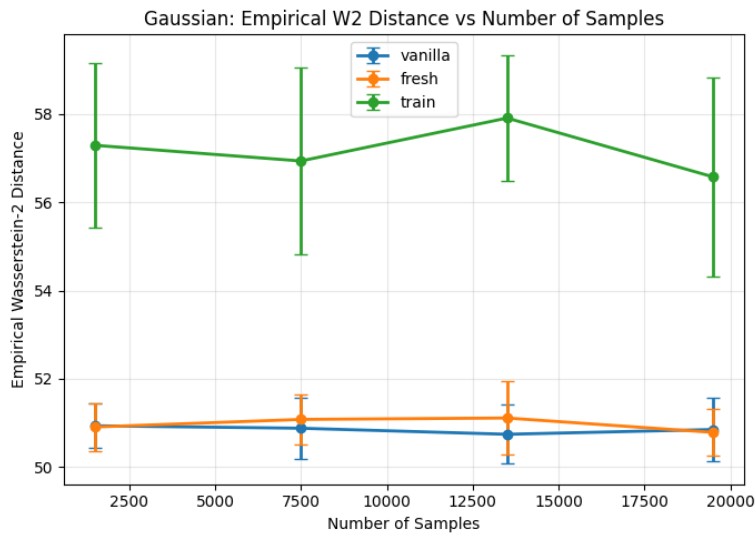

*Figure 4.* Empirical Wasserstein distance of produced samples vs Gaussian $\pi_{\text{tar}}$.

Figure 4 shows a similar result to Figure 2. For Gaussian $\pi_{\text{tar}}$, Algorithm 3 (initialize at some of the training samples) performs significantly worse than Algorithm 2 (initialize at fresh samples) and Algorithm 1 (standard normal initialization).

