# OpenReview forum: "On the Robustness of Langevin Dynamics to Score Function Error"
_ICML.cc/2026/Conference — ICML 2026 regular_

### Official Review · Reviewer_HfsE · 2026-02-26

**Soundness:** 2
**Presentation:** 1
**Significance:** 2
**Originality:** 3
**Overall Recommendation:** 3
**Confidence:** 4

**Summary:**

This paper studies the robustness of Langevin Dynamics (LD) to score estimation errors, with the central question being whether a small $L^2$ error in a learned score function is sufficient for accurate sampling in high dimensions. The paper contrasts this with the diffusion-model setting, where robustness guarantees are known under suitable control of the score error across the annealing path.

The main message of the paper is negative for vanilla LD: the authors show that even an exponentially small $L^2$ score error does not, by itself, guarantee convergence to the target distribution. The paper presents three main theoretical results:

1. **Standard initialization (Theorem 3.1):** For a Gaussian target, the authors construct a score estimate that has very small $L^2$ error under the target distribution but is misleading in the region where the chain is initialized, causing the LD dynamics to become trapped.

2. **Data-based initialization (Theorem 3.7):** When LD is initialized from the training data, the paper shows that an overfitted score estimator can create local traps around memorized data points, preventing proper mixing.

3. **General targets (Theorem 3.11):** For more general target distributions, the authors construct an adversarial “cone trap” example showing asymptotic failure of LD as $t \to \infty$.

The paper also includes experiments on Gaussian and Gaussian-mixture targets aimed at illustrating the practical implications of the data-based initialization setting. These experiments suggest that, in an over-parameterized regime, initializing LD from fresh samples may outperform initialization from the training data itself.

**Compliance With Llm Reviewing Policy:**

Affirmed.

**Final Justification:**

After considering both the paper and the authors’ rebuttal, I am updating my overall recommendation from 2 (Reject) to 3 (Weak Reject).

The rebuttal addressed a meaningful part of my original concerns. In particular, the clarification that Theorem 3.7 is the main result, and that Section 4 is intended primarily to support Theorem 3.7, helped resolve my earlier concern about the theory-experiment mismatch. I also found the additional simulations for Theorem 3.1 helpful, as well as the commitment to include a more systematic study over $\\|\mu\\|$ in a revision. These points make the empirical and conceptual narrative of the paper clearer than it was in the original submission.

In terms of strengths, I continue to find the paper technically interesting and, overall, mathematically careful. The main question it studies is important, and I still find the data-based initialization result in Theorem 3.7 the most original and practically compelling contribution.

However, I still have some remaining reservations. My main concern is about the scope and practical interpretation of the broader negative message beyond Theorem 3.7. While I now better understand that Theorems 3.1 and 3.11 are intended as examples showing that small $L^2$ score error alone is insufficient, I remain less convinced about the practical relevance of Theorem 3.11 in particular. The cone-trap construction appears mathematically valid, but it is still unclear to me whether it should be viewed mainly as a worst-case impossibility-style result or as a mechanism that is likely to arise in realistic learned-score settings. Since this point was not fully clarified in the rebuttal, it continues to limit my assessment of the paper’s overall significance.

Overall, the rebuttal improved my view of the paper, especially by clarifying the role of Theorem 3.7 and by strengthening support for Theorem 3.1. However, I still think the paper would benefit from sharper framing of the hierarchy and practical scope of its results, especially with respect to Theorem 3.11. For this reason, my final recommendation is 3 (Weak Reject).

**Key Questions For Authors:**

### Q1. On the theory-experiment connection for Theorem 3.1

In the Gaussian setting, Theorem 3.1 considers a target $N(\mu, I_d)$ with $\|\mu\| = 7\sqrt d$, which places the target far from the standard initialization $N(0, I_d)$ in high dimension. By contrast, in the Gaussian experiment in Section 4, the target appears to be much closer to the initialization distribution (roughly on the order of $\sqrt d$ in $d=50$). Because of this, I found it somewhat difficult to assess how directly the experiment is meant to speak to the failure mode in Theorem 3.1.

- **Question:** Why was this particular experimental regime chosen? If the authors intend the Gaussian experiment to provide empirical support for Theorem 3.1, could they provide additional experiments (or commit to including them in a revision) in which $\|\mu\|$ is varied more systematically, especially in a regime closer to $\|\mu\| \approx 7\sqrt d$?

- **Why this matters for my evaluation:**
At present, I find the experiments most convincing as support for the data-based initialization result. If the authors can either clarify that this is the intended scope of Section 4 or provide additional evidence connecting the experiments more directly to Theorem 3.1, it would strengthen my view of the empirical support for the paper’s broader narrative.

### Q2. On the interpretation of Theorem 3.1

My current reading is that the main mechanism in Theorem 3.1 is that the chain is initialized in a region where the target density is negligible, so the $L^2(\pi_{\mathrm{tar}})$ score-matching objective does not meaningfully constrain the learned score field there. In that sense, the result seems to show most directly that small $L^2$ score error does not control extrapolation behavior outside the data region.

- **Question:** Do the authors view Theorem 3.1 primarily as an extrapolation-type limitation of the standard $L^2$ score-matching objective, or as evidence of a broader lack of robustness of vanilla LD in practically typical settings?

- **Why this matters for my evaluation:**
This distinction is important for how I assess the significance of the theorem. If the intended takeaway is mainly about the lack of control in low-density regions, I would interpret the result as interesting but somewhat narrower in scope. If the authors can argue that the same mechanism is likely to matter more broadly in practice, that would strengthen my assessment of the result’s practical significance.

### Q3. On the practical relevance of Theorem 3.11

Theorem 3.11 gives a strong general negative result via the construction of a “cone trap” extending to infinity. I understand the mathematical value of such a worst-case example, but I was less sure how representative this construction is of approximation errors that might arise in learned neural score models trained on finite data.

- **Question:** Do the authors have either empirical evidence or theoretical reasons to believe that similar large-scale trap structures can arise naturally in practical score-learning settings? If not, would they consider framing Theorem 3.11 more explicitly as a worst-case impossibility result?

- **Why this matters for my evaluation:**
This would help me better judge the practical significance of Theorem 3.11. If the construction is mainly intended as a worst-case counterexample, I would still view it as mathematically interesting, but I would interpret its practical scope more narrowly. A clearer connection to realistic learned-score behavior would strengthen my assessment of this part of the paper.

### Q4. On presentation and experimental balance

I felt that the introduction and summary of results occupy substantial space, while the experimental section is relatively brief. Since some of the key theoretical constructions are geometric in nature, I wondered whether the paper might benefit from additional empirical or visual support.

- **Question:** Would the authors be open to shortening the introduction / summary section and using that space for either additional ablations (for example, over the separation parameter $\|\mu\|$) or schematic illustrations of the trapping geometries in Theorems 3.1 and 3.11?

- **Why this matters for my evaluation:**
This would not change my view of the core theory, but it could improve both the presentation and my confidence in the practical interpretation of the paper.

**Limitations:**

The paper would benefit from a more explicit discussion of the limitations of its theoretical results. While the authors describe their examples as “natural,” some of the constructions rely on specific geometric mechanisms that may not fully reflect the average-case behavior of neural networks trained with score matching. I would encourage the authors to add a short paragraph (for example, in the conclusion) discussing the following points:

- **Scope of the theoretical constructions:** The failure modes in Theorems 3.1 and 3.11 rely on specific structures (e.g., large separation between initialization and target, or an infinite cone trap). It would be helpful to clarify whether these are intended as realistic failure modes in practice or primarily as worst-case counterexamples.

- **Nature of the failure in Theorem 3.1:** It would help to state more explicitly that the failure mechanism in Theorem 3.1 is tied to initialization in low-density / extrapolation regions, where small $L^2$ error under the target distribution provides limited control over the learned score field. Clarifying this would make the scope of the contribution easier to interpret.

- **Experimental limitations:** Since the experimental section appears most directly connected to the data-based initialization setting, it would be useful to discuss the extent to which the experiments support the broader theoretical narrative, especially for the standard-initialization setting.

I do not see any immediate negative societal impact beyond the usual considerations associated with generative modeling.

**Strengths And Weaknesses:**

## Strengths

### Soundness
- The theoretical results appear technically careful and nontrivial. In particular, the trapping constructions, concentration-based arguments, and comparison-style reasoning used to analyze the Langevin dynamics seem rigorous to me.
- The paper is also fairly clear about what kind of guarantee it is ruling out: namely, whether small $L^2$ score error alone is sufficient to ensure reliable sampling with vanilla Langevin Dynamics.

### Presentation
- The paper is generally readable, and the main claims are stated clearly. I was able to follow the high-level message of each theorem without too much difficulty.
- The progression from the standard-initialization setting to the data-initialization setting and then to more general targets gives the paper a coherent overall structure.

### Significance
- I think the paper studies an important question in score-based generative modeling: whether accurate score estimation in the usual $L^2$ sense is enough to guarantee successful sampling.
- This is a meaningful question both theoretically and practically, since Langevin Dynamics and learned score fields are central objects in the modern generative-modeling literature.
- In particular, the paper offers a useful cautionary perspective by showing that guarantees that hold in annealed settings do not automatically transfer to vanilla LD.

### Originality
- The overall perspective is original, especially in the way the paper frames robustness of LD as a question about the adequacy of the score-matching objective.
- I found the data-based initialization result particularly interesting. The idea that an overfitted score estimator can create local traps around memorized samples provides a novel and intuitive connection between overfitting and sampling failure.

## Weaknesses

### Soundness
- The theoretical arguments appear technically careful overall, but I have some reservations about how broadly the negative results should be interpreted. In Theorem 3.1, the construction considers initialization from $N(0,I_d)$ and a target $N(\mu,I_d)$ with $\|\mu\|=7\sqrt d$, so that in high dimension the initialization is effectively far from the high-density region of the target. My reading is that the result shows, quite convincingly, that small $L^2(\pi_{\mathrm{tar}})$ score error does not constrain the learned score field in such low-density / extrapolation regions. This is an important limitation, but I am less certain that it should be interpreted as a broad lack of robustness of LD in more typical regimes.
- Similarly, Theorem 3.11 is mathematically valid as a worst-case construction, but the “cone trap” extending to infinity felt somewhat artificial to me. I would have appreciated more discussion of whether such large-scale coherent trap structures are expected to arise from realistic approximation errors in learned neural score networks.

### Soundness / Significance
- I also found the connection between the theory and experiments somewhat limited. Theorem 3.1 uses a separation scale $\|\mu\|=7\sqrt d$, whereas in the Gaussian experiment the target mean is much closer to the initialization distribution (roughly on the order of $\sqrt d$ in $d=50$). Because of this, the experimental setup does not seem to directly test the standard-initialization failure mode proved in Theorem 3.1. I do not view this as a contradiction, since Section 4 seems more closely tied to Theorem 3.7, but I think the empirical support for the broader narrative would be stronger if the paper varied $\|\mu\|$ more systematically.

### Presentation
- I felt that the structure of the paper is somewhat unbalanced. The introduction and summary of results occupy substantial space, while the experiments are relatively brief. I would have preferred either more empirical ablations—for example, over the separation parameter $\|\mu\|$—or simple schematic figures illustrating the trapping geometries in Theorems 3.1 and 3.11.

---

> ### Author Rebuttal · Authors · 2026-03-30
>
> Dear reviewer, thank you for your careful reading and thoughtful feedback. We greatly appreciate your engagement with our manuscript and are glad you found that “the paper studies an important question in score-based generative modeling” and that you “found the data-based initialization result particularly interesting”. We will be sure to take your suggestions fully into account to improve notation, presentation, and clarity of our manuscript.
>
> Overall, we emphasize that *Theorem 3.7 on data-based initialization is our main result*. We also emphasize that the simulations in Section 4 are intended to provide support only for Theorem 3.7.
>
> We would like to respond to the specific weaknesses and questions you raised below, and we believe this will help clarify the results of our manuscript.
>
> **On practical relevance of Theorems 3.1 and 3.11:** We view Theorems 3.1 and 3.11 as constructing particular examples demonstrating that $L^2$-accuracy alone is insufficient for the success of Langevin dynamics, in contrast to diffusion models.
>
> We instead view Theorem 3.7 as our main result, establishing how this lack of $L^2$ robustness can also arise in more practical settings with an overfit score estimator, as validated in synthetic simulations in Section 4. We also provide additional empirical evidence that the local traps described in Theorem 3.7 can arise from overfitting in learned score estimators; please see our additional simulations in our response to Reviewer MgRM.
>
> **On the simulations vs Theorem 3.1**: We clarify that the current simulations in Section 4 are meant to support the data-based initialization result, Theorem 3.7.
>
> We now provide additional simulations supporting Theorem 3.1. We construct a target $N(\mu, I_d)$ with $\||\mu\||=7\sqrt{d}$ and a perturbed score as considered in Theorem 3.1. The $L^2$ error of the perturbed score is miniscule – an empirical estimate based on i.i.d. samples from the target is 0 up to 10 significant digits.
>
> We then sample using Langevin dynamics with this perturbed score versus the true score, both initialized at $N(0, I_d)$. We compute the KL and sliced 2-Wasserstein distance of the samples to the target and find that the performance of Langevin with the perturbed score is significantly worse than with the true score. Here KL is computed by fitting outputs to a Gaussian and using the analytical formula for KL between Gaussians. Results are averaged and errors are reported over 10 trials.
>
> These results empirically confirm that even with a tiny $L^2$ error, initialization far from the target can still lead to substantial sampling failure, consistent with Theorem 3.1. We commit to expanding these simulations in support of Theorem 3.1 in a revision of the manuscript, where we will systematically vary $\|| \mu\||$.
>
> | | Sampling with perturbed score | Sampling with true score |
> |---|---|---|
> | KL to target | $30597.5402 \pm 560.2735$ | $8.4426 \pm 0.1637$ |
> | Sliced 2-Wasserstein distance to target | $7.0053 \pm 0.1250$ | $0.5780 \pm 0.0115$ |
>
>
> **On presentation and experimental balance:** We thank the reviewer for these valuable suggestions on presentation. In a revision we will shorten the introduction/summary section, add additional experimental results in the main body, and add schematic illustrations of the geometric constructions of Theorems 3.1, 3.7, and 3.11. We will also add a short paragraph in the conclusion qualifying the scope and nature of our results, as per your suggestion in Limitations.
>
> We hope these clarifications address your concerns and clarify the scope and practical relevance of our results. We greatly appreciate your detailed comments and are very open to further discussion. Thank you again for your time and feedback!

---

> > ### Author Rebuttal · Reviewer_HfsE · 2026-04-02
> >
> > Thank you for the detailed rebuttal. My concerns are partially resolved.
> >
> > I appreciate the clarification that Theorem 3.7 is the main result and that Section 4 is intended primarily to support Theorem 3.7. The additional simulations for Theorem 3.1 and the commitment to include a more systematic study over $\\|\mu\\|$ in a revision are also helpful. I also appreciate the additional evidence that overfitted score estimators can indeed induce the trapping behavior described in Theorem 3.7.
> >
> > My remaining concern is mainly about the scope and practical relevance of Theorem 3.11. While I confirmed the additional experimental results on the answer for Reviewer MgRM, I am still unsure from the current response how directly the practical relevance of the cone-trap construction has been established. Could the authors clarify whether Theorem 3.11 is primarily intended as a worst-case / impossibility-style result, or whether they believe similar structures can arise naturally in realistic learned-score settings?
> >
> > Could the authors clarify, in the revision, whether they view Theorem 3.11 primarily as a worst-case / impossibility-style result, or whether they believe similar large-scale trap structures can arise naturally in realistic learned-score settings? A clearer positioning of this theorem would help me assess its practical significance.

---

> > > ### Author Response · Authors · 2026-04-07
> > >
> > > We thank the reviewer for the helpful follow-up, and we are glad the reviewer’s concerns are partially resolved. We understand that the reviewer has already posted their final justification, but we hope we can give additional context and that the reviewer will find our comment useful.
> > >
> > > Regarding Theorem 3.11: it is intended as a worst-case impossibility style result. Its purpose is to show that the lack of robustness of Langevin dynamics to $L^2$ score-estimation error is not specific to Gaussians (Theorems 3.1, 3.7) or strongly-log-concave measures (Remarks 3.5, 3.9).
> > >
> > > We agree that clearer positioning of Theorem 3.11 would improve its practical interpretation and we will be sure to position it more clearly in a revision.
> > >
> > > We reiterate that Theorem 3.7 is our main result. Our simulations in the manuscript and in reply to reviewer MgRM confirm that in synthetic settings, the trap behavior in Theorem 3.7 can arise naturally in learned-score settings.
> > >
> > > Thank you again for the continued engagement and constructive feedback!

---

### Official Review · Reviewer_3NzB · 2026-03-04

**Soundness:** 3
**Presentation:** 4
**Significance:** 2
**Originality:** 2
**Overall Recommendation:** 4
**Confidence:** 4

**Summary:**

The paper proposes to study the properties of Langevin dynamics under an $L^2$ approximation of the score in high-dimensions. The main theoretical argument that is present throughout the paper is the presentation of what one might call "ill-behaved" examples, with increasing degree of "reality". Namely, examples where the authors propose a score approximation $\tilde{s}$ for a given score distribution $\pi$ with score $s$ such that $\mathbb{E}_{\pi}[\|s(X) - \tilde{s}(X)\|^2]$ can be made rendered as small but nevertheless sampling according to a Langevin dynamics with "ill-behaved" initialization will behave poorly, namely by taking an exponential time to achieve convergence over the total variation distance.

The paper is concluded by a brief numerical section in toy examples where learning of the score uses a Neural Network and the now well known Denoising score matching objective.

**Compliance With Llm Reviewing Policy:**

Affirmed.

**Final Justification:**

The authors have provided more evidence of the "trapped in a cone" relevance in practical examples and addressed most of my comments. I still feel that the fact the authors used a DDPM approach for learning the score unnecessary and I have some doubts over the impact of the paper results in practice, but it is a well written paper with new results with clear and rigorous proofs, thus I am willing to increase my score to weak accept as I feel that the overall quality out-weight the flaws mentioned above.

**Key Questions For Authors:**

Key Questions:
1. Why did the authors adopt the current learning method? Namely, why not learn only a single level of noise?
2. How key is the fact of using a limited dataset of $1000$ (repeated) points important? Surely it is a poor monte carlo approximation of the objective, so actually understanding the impact of the number of points key.
3. Is the noise, during the denoising score matching learning phase, added "dynamically" or statically? Namely, for each batch a new noise is added to the base samples from $\pi$ or the same noise is used throughout the batches?
4. Why compare the noisy learned score to $\pi$ and not the score of the noisy distribution? If comparison with $\pi$ is needed, one should try another learning scheme equivalent to $L2$ minimization tool such as implicit score matching or sliced score matching.
5. Line 1029: "Since $Y_t \rightarrow \pi_{tar}$, we have $P(T_1^Y < \infty) =1$" Is this true? I think it might be true for $Y_t$ such as it is defined but it is not enough that $Y_t \rightarrow \pi_{tar}$, an ergodicity condition should also be required. Indeed, taking $Y_0 \sim \pi_{tar}$ and setting $Y_t = Y_0$ for all $t$ we would have $Y_t \rightarrow \pi_{tar}$ and  $P(T_1^Y < \infty) < 0$

Other questions and typos:
1. I think there is a $\sup_t$ missing in a lot of places in the proof of theorem A.T, namely lines 668 but also $696$.
2. In the array of equations from line 682 to 685 can't the inequality be enhanced by using the fact that the brownian is a Martingale?
3. Lines 691 and 692: I believe that for the second space, the covering number should be $\mathcal{N}([0,1], d_2, \epsilon/6)$ instead of $\mathcal{N}(S^{d-1}, d_2, \epsilon/6)$.
4. Begining of the proof of lemma 3.4: It believe it should be $\hat{s}(X_t(x_0)) = - \alpha X_t(x_0)$ instead of $\hat{s}(x) = - \alpha x$.
5. Equation array starting on line 975: I believe it should be an intersection of the cone and the level sets instead of an union.

**Limitations:**

Yes

**Strengths And Weaknesses:**

* Soundness: The paper is technically sound. Although I spotted some minor issues in the theoretical part (see below), I believe most that all of the theorems should be true and the proof techniques are on par with the current littérature on the tools that the authors use (as far as I am concerned). However, I am afraid the same level of rigour that is used for the theoretical part is lacking in the numerical part. Several choices of the numerical part seem poorly justified. I'll pinpoint those points further in the question part of the review, but it is not clear why using the so called standard "DDPM" learning instead of learning the score for a single noise level ? Furthermore, for the GMM case the confidence intervals completely overlap without even mention the fact that the empirical Wasserstein distance is known to behave extremely badly in high dimensions.

* Presentation: The submission is clearly written, both the main and the technical parts and I have no comments in this part.

* Significance and Originality: This is the point of main concern in my opinion. The fact that running Langevin of an $L^2$ approximated score for a noisy version of the distribution might behave badly is well known in the literature (at least empirically). This is precisely the point made by the authors of [1] in figure 2 and section 3.2.1. However, as far as I am concerned the authors are the first to precisely quantify such phenomenon and to show mathematically, which is of course of some significancy. However, the main theoretical tools for such analysis are somehow well known. So I am actually divided between the fact that I think it is of some importance to actually put the work of assembling the well known fact that empirically one should not use Langevin sampler with an L2 score, specially if one has no idea how to initialize the distribution, and knowledge in mathematical statistics and stochastic processus theory to establish lower bounds on the total variation distance and the fact that it was somehow considered a well known fact, indeed discussed in one of the founding papers of what is now known as score-based generative modelling.


[1] Song, Y., & Ermon, S. (2019). Generative modeling by estimating gradients of the data distribution. Advances in neural information processing systems, 32.

---

> ### Author Rebuttal · Authors · 2026-03-30
>
> Dear reviewer, thank you for your careful reading and thoughtful review. We are glad you found the paper well-written, and we greatly appreciate the time and effort you put into providing detailed and constructive feedback. We will be sure to incorporate your suggestions, including fixing the typos you identified in the appendix.
>
> We respond to your questions and concerns below, and we believe this will help clarify several of the results and choices from the manuscript. All sources here are the same as in our manuscript or are given below.
>
> **Choices made in simulations**: We would like to clarify our design choices as follows. Our goal in the simulations was to use the learned score to sample from $\pi$, with the specific aim of demonstrating the practical implication of Theorem 3.7: when the score estimate is overfit to the training samples, Langevin dynamics can fail when initialized from these same points. To achieve this, our goal in training a score estimate was to learn an overfit approximation of $\nabla \log \pi$.
>
> 1. *Learning method:* We used the DDPM objective with low noise levels because it yielded the most accurate and stable approximation of $\nabla \log \pi$. Using a single, very low noise level proved difficult to train reliably. While methods such as implicit score matching or sliced score matching could in principle be used, they are less practical even at small scale. We chose DDPM with low noise levels because it is fast, easy to implement, and sufficient to illustrate the phenomenon highlighted by Theorem 3.7.
>
> 2. *Limited dataset of 1000 points:* We intentionally used a limited dataset of 1000 repeated points to encourage overfitting of the score function to these training points, and therefore demonstrate the phenomenon predicted by the theorem.
>
> 3. *On the addition of noise:* During training, we added noise ‘dynamically’: for each batch, new noise was sampled and added to the base samples from $\pi$, following the standard DDPM procedure.
>
> We will be sure to revise Section 4 to make these choices and their motivations more explicit.
>
> **Significance of our results:** Our results carry further practical significance by highlighting the crucial role of using fresh samples in data-based sampling methods: Theorem 3.7 asserts that Langevin dynamics fails to converge when initialized at the *same* samples used to learn an $L^2$-accurate score. Our simulations confirm this in synthetic settings. In contrast, prior literature (Koehler et al., 2024/2025) established Langevin dynamics succeeds under $L^2$ accuracy conditions when initialized at *fresh* samples.
>
> Taken together, these results provide clear guidance and novel insights. In practice, to use fresh samples properly when using data-based initialization or related approaches (such as energy-based models and contrastive divergence), one should separate the training set into a group for learning the score function and a hold-out set for initialization. This is a simple but important consideration for practical implementations that can be easily overlooked.
>
> **Regarding line 1029:** We thank the reviewer for the thorough reading and noticing this subtlety. To rigorously justify this step, it suffices that 1) the density of $\pi_{\text{tar}}$ is strictly positive w.r.t. Lebesgue measure and 2) $\log \pi_{\text{tar}}$ satisfies a dissipativity condition (as commonly assumed in the literature [1]). Combined with Assumption 3.10 (Lipschitzness of $\nabla \log \pi_{\text{tar}}$), the associated diffusion (from lines 1025-26) is non-explosive, irreducible, and positive Harris recurrent with invariant distribution $\pi_{\text{tar}}$. Since $\pi_{\text{tar}}(K’)>0$ by 1) as $K’$ has strictly positive Lebesgue measure, we obtain $P(T_1^Y<\infty)=1$.
>
> We hope these clarifications help address your concerns. We greatly appreciate your comments and are very open to further discussion. Thank you again for your time and feedback!
>
> [1] Maxim Raginsky, Alexander Rakhlin, Matus Telgarsky. "Non-Convex Learning via Stochastic Gradient Langevin Dynamics: A Nonasymptotic Analysis." COLT 2017

---

> > ### Author Rebuttal · Reviewer_3NzB · 2026-04-02
> >
> > I thank the authors for the answer, however I still have a few points that I'd like to bring to the attention of the authors.
> >
> > First of all, my point is that even though using the "DDPM" approach to learn the score, one should still compare what is comparable. Thus, as DDPM learns the score of a noisy version of $\pi$, the authors should compare with samples from this noisy version, and not with the "clean" $\pi$. That was the point of my question 4.
> >
> > I understand the fact that the goal was to overfit, but still I'd like to understand if the "trapped in a cone" continues to exist when  the score is "less overfitted", in line with the doubts that reviewer "HfsE" raised in his review and review acknowledgment.

---

> > > ### Author Response · Authors · 2026-04-07
> > >
> > > We thank the reviewer for the continued engagement and constructive feedback.
> > >
> > > Regarding DDPM, we agree that it learns the score of a lightly noised version of $\pi$. Our comparison to the clean $\pi$ was intentional, since the goal of Section 4 was to use a practical score-learning procedure to obtain an overfit score estimate and then study Langevin sampling for the underlying clean target. We acknowledge the reviewer’s point, and we will clarify this motivation more explicitly in a revision.
> > >
> > > Regarding whether trapping persists with less overfitting: we agree it would be valuable to study how this behavior changes in milder overfitting regimes, and we commit to including such ablations in a revision. Our current simulations are designed specifically to illustrate the heavily overfit regime studied in Theorem 3.7, where the effect is indeed strongest. We note that additional simulations supporting this effect with overfit, learned scores are in our reply to Reviewer MgRM.
> > >
> > > Thank you again for the continued engagement and constructive feedback!

---

### Official Review · Reviewer_bAgU · 2026-03-12

**Soundness:** 3
**Presentation:** 4
**Significance:** 2
**Originality:** 3
**Overall Recommendation:** 5
**Confidence:** 3

**Summary:**

The paper studies the robustness of the Langevin algorithm under an $L^2$ bound on the score error.

In a first theorem, the authors show that, in a special case where the $L^2$ bound is of order $\exp(-Cd)$, the Langevin algorithm does not converge within a time that is exponential in the dimension.

This result confirms that the bound of Theorem 2.1 in [1] is sharp, and that one needs an $L^2$ bound of order $\varepsilon \exp(-cd)$ (for an explicit $c$ depending on the initial measure) to achieve a total variation error of order $\varepsilon$ after a time polynomial in $d$.

The authors also present another result where the initial measure of the Langevin algorithm is the empirical measure of $n$ i.i.d. samples $(x^i)$ from the target distribution. They show that if the score matching error depends on the samples $x^i$, the algorithm does not converge in any time polynomial in $d$.

In a more general setting, for any strictly positive score error $\varepsilon_s$ and a target distribution satisfying mild assumptions, there exists a score function with error $\varepsilon_s$ that makes the algorithm diverge from the target as $t \to \infty$.



[1]: Lee, Holden, Jianfeng Lu, and Yixin Tan. "Convergence for score-based generative modeling with polynomial complexity." Advances in Neural Information Processing Systems 35 (2022): 22870-22882.

**Compliance With Llm Reviewing Policy:**

Affirmed.

**Final Justification:**

The authors propose a well-written paper that highlights an important issue, namely that a score error of $ e^{-d} $ is not sufficient, and that the Langevin algorithm should be initialized with fresh samples.

My only concern is the significance of the contribution, since the authors mainly confirm an intuition from prior work: that in order to obtain theoretical guarantees, the score error must be smaller than $e^{-d}$.

The authors address my concerns in the rebuttal. If the paper is accepted, they should revise the definition (as stated in the rebuttal) of $\hat{s}$ in the theorem.

**Key Questions For Authors:**

- 1: What is $x^i$ in Theorem 3.7? How do you pass from the initialization $X_0 = x^i$ to $X_0 = n^{-1}\sum_{I \in [n]} \delta_{x^I}$ in line 378?

- 2: How do you explain that Algorithm 2 does not converge in Figure (a), as this seems to contradict Theorem 26 in [2]?

- 3: In your numerical experiments, how does the $L^2$ error of the score behave? In particular, how can one check whether the assumptions of your theorems are satisfied in these experiments?

I would be happy to raise my score if you could clarify Theorem 3.7 and the results of the numerical experiments.


[2]: Koehler, Frederic, Holden Lee, and Thuy-Duong Vuong. "Efficiently learning and sampling multimodal distributions with data-based initialization." The Thirty Eighth Annual Conference on Learning Theory. PMLR, 2025.

**Limitations:**

yes

**Strengths And Weaknesses:**

**Strengths:**
- The paper is well written and the contributions are clearly presented.
- The paper establishes an intuitive counterexample showing that the Langevin algorithm does not converge in polynomial time in $d$ when the score error is only of order $e^{-d}$, in the case where the initial distribution is far from the target.

**Weaknesses:**
- Theorem 3.1 mainly confirms an intuition from previous work. In particular, [1] provides a convergence result where the $L^2$ score error must be of the form $\varepsilon e^{-d}$, which already suggests that an $L^2$ score error of $e^{-d}$ alone is not sufficient.
- Theorem 3.7 is unclear because $x_i$ is not defined. In my understanding, Equation (9) should be initialized with $X^0 = x_i$, according to the proof sketch. However, in that case the message of the theorem becomes weaker.
- In the simulations, it is surprising that Algorithm 2 does not converge, since it is known to converge according to Theorem 26 in [2].
- Minor remark: in Theorem 3.11, I do not see the purpose of introducing $\delta_{\theta,u}$.

[1]: Lee, Holden, Jianfeng Lu, and Yixin Tan. "Convergence for score-based generative modeling with polynomial complexity." Advances in Neural Information Processing Systems 35 (2022): 22870-22882.

[2]: Koehler, Frederic, Holden Lee, and Thuy-Duong Vuong. "Efficiently learning and sampling multimodal distributions with data-based initialization." The Thirty Eighth Annual Conference on Learning Theory. PMLR, 2025.

---

> ### Author Rebuttal · Authors · 2026-03-30
>
> Dear reviewer, thank you for your careful reading and thoughtful and supportive review. We are glad you found the paper clearly written, and we appreciate your insightful questions. We answer the questions and weaknesses you raised as follows. All sources below are the same as in our manuscript or your review.
>
> **Initialization and the proof sketch of Theorem 3.7:** This is a good question, and we thank the reviewer for raising it. To answer: Theorem 3.7 is stated with initialization $\frac1n \sum_i \delta_{x_i}$. Here, $x_i$ in the proof sketch denotes an arbitrary support point of this empirical initialization (i.e., any of the data points $x_1, \ldots, x_n$), and the argument analyzes the law of Langevin dynamics conditioned on initializing at $x_i$.
>
> In particular, for any $x_i \in \\{x_1, \ldots, x_n\\}$, Langevin dynamics fails to escape a neighborhood of $x_i$ on sub-exponential time scales (provided the $x_i$ are in ‘General Position’, Def 3.6, which occurs with probability $1-e^{-\Omega(d)}$). Since the argument holds uniformly over all $x_i$, averaging over the empirical initialization in Eq. (9) recovers the Theorem (formalized in Appendix B.2).
> We will be sure to revise Theorem 3.7 and the proof sketch to make this clear, and we thank the reviewer for bringing up this point.
>
> **Consistency with results of Koehler, Lee, and Vuong [2]:** Our results are fully consistent with those of [2]. In Figure (a), the algorithm that performs worst is Algorithm 3 ('train', green line), which is initialized at the *same* samples used for learning the score function, consistent with Theorem 3.7. In contrast, [2] (and also Koehler and Vuong 2024) considers initialization at *fresh* samples that are not used to learn the score; this corresponds to Algorithm 2 in Figure (a) ('fresh', orange line), which performs substantially better. We will revise the Figure (a) caption and the discussion in Section 4 to make the distinction between Algorithm 2 and 3 clearer.
>
> We would also like to note this distinction between fresh samples versus reused training samples is practically relevant. Taken together with [2], our results suggest that fresh samples are critical for the efficacy of data-based initialization (and related approaches such as contrastive divergence and energy-based models).
>
> **Verifying assumptions in simulations:** We thank the reviewer for asking this important question. We do not claim that the simulations verify every assumption of Theorem 3.7 exactly. Rather, they are designed to test its qualitative prediction: when the score estimate is overfit to the training samples, Langevin dynamics can fail when initialized from these same training samples.
>
> To test this, we trained the score network for a very large number of iterations (150,000) from a limited set of training samples. We consistently observed relatively small training $L^2$ error (~0.46 for  Gaussian target, ~0.14 for GMM), while the $L^2$ error on fresh samples remained substantially larger, consistent with overfitting.
> The goal of Section 4 is therefore not a literal instantiation of the Theorem’s assumptions, but to verify its qualitative predictions in synthetic settings.
> We will be sure to clarify this goal in a revision of the manuscript.
>
> We hope this resolves the ambiguity in Theorem 3.7 and clarifies why the simulations are consistent with prior literature. We will revise the discussion and proof sketch of Theorem 3.7 as well as Section 4 accordingly. Thank you again for your time and feedback!

---

> > ### Author Rebuttal · Reviewer_bAgU · 2026-04-03
> >
> > Thank you for your answers.
> >
> > However, I am sorry to say that Theorem 3.7 remains unclear to me. In particular, $\hat{s}$ is not well defined, and the index $i$ is not specified. Is the statement intended to hold for every $i$, or only for $i = 1$? As currently stated, I am unable to follow the proof of the theorem.
> >
> > Could you please provide an explicit definition of $\hat{s}$?

---

> > > ### Author Response · Authors · 2026-04-07
> > >
> > > We thank the reviewer for their continued engagement and the very helpful question. Yes, *Theorem 3.7 is intended to hold for every $i$*. We now provide an explicit definition of $\hat s$.
> > >
> > > Let $x_1, \ldots, x_n \sim \pi_{\text{tar}}$ be i.i.d., and suppose $x_1, \ldots, x_n$ are in general position (Def 3.6).
> > >
> > > Crucially, by the discussion immediately preceding Theorem 3.7, the fact that $x_1, \ldots, x_n$ are in general position implies for all $x \in \mathbb{R}^d$, exactly one of the following holds:
> > >
> > > 1. $\|x - x_i \| \ge 0.16\sqrt{d}$ for all $i, 1 \le i \le n$.
> > >
> > > 2. Or there is a *unique* $i \in \{1, 2, \ldots, n\}$ such that $\|x - x_i\| \le 0.16\sqrt{d}$.
> > >
> > > This uniqueness ensures that $\hat s$ is well defined.
> > >
> > > The definition of $\hat s$ as written in Theorem 3.7 now makes sense: for any $x \in \mathbb{R}^d$, we define $\hat s(x)$ by:
> > >
> > > 1. If for all $i, 1 \le i \le n$, $\|x - x_i \| \ge 0.16\sqrt{d}$, set $\hat s(x)=-x$.
> > >
> > > 2. Otherwise, we let $i \in \{1, 2, \ldots, n\}$ be the *unique* index such that $\|x - x_i\| \le 0.16\sqrt{d}$. Then:
> > >
> > >     If $\|x-x_i\| \le 0.15\sqrt{d}$, set $\hat s(x) = -\alpha (x-x_i)$.
> > >
> > >     Otherwise if $0.15\sqrt{d} < \|x-x_i\| < 0.16\sqrt{d}$, set $\hat s(x) = -\psi\big( \frac{100(x-x_i)-11}{\sqrt{d}} \big) \cdot \alpha (x-x_i) - \big(1-\psi\big( \frac{100(x-x_i)-11}{\sqrt{d}} \big) \big) \cdot x$.
> > >
> > > The remainder of Theorem 3.7 then proceeds exactly as written.
> > >
> > > In a revision, we will make this definition and the well-definedness of $\hat s$ fully explicit in the statement of Theorem 3.7. We again thank the reviewer for the engagement and hope our response fully clarifies Theorem 3.7.

---

### Official Review · Reviewer_MgRM · 2026-03-13

**Soundness:** 3
**Presentation:** 3
**Significance:** 3
**Originality:** 3
**Overall Recommendation:** 5
**Confidence:** 3

**Summary:**

This work study a series of valuable count examples for the sampling process of Langevin and show the different property between the Langevin dynamic and diffusion models. More specifically, this paper construct explicit examples showing that even when the $L^p$ score estimation error is exponentially small in dimension, Langevin dynamics can fail catastrophically in high dimensions even in a easy setting. These results stand in sharp contrast to diffusion models, where $L^2$ score estimation error suffices for polynomial-time convergence, providing fundamental insight into the theoretical gap between Langevin dynamics and diffusion models.

**Compliance With Llm Reviewing Policy:**

Affirmed.

**Key Questions For Authors:**

1.	The score estimates in the main theory are constructed by carefully modifying the true score in a specific annular region using smooth bump functions. While the authors emphasize that these estimates are Lipschitz with small estimation error, it is unclear whether score matching with neural networks would naturally produce such pathological behavior in low-probability regions.

**Limitations:**

yes

**Strengths And Weaknesses:**

Strength:

1.	This work directly faces a important question: whether L^2 score estimation suffices for sampling and give a negative answer for Langevin dynamic, which is valuable.

2.	The idea of exploiting high-dimensional concentration to "hide" score errors in low-probability regions of target distribution is clever and elegant. The reduction to escape-time analysis of an OU process and the cone-exit analysis for Brownian motion with drift are interesting technical contributions.

3.	The simulation experiments also supports the theoretical guarantee.

Weakness:

1.	As a series of works of diffusion models use the underdamped Langevin as a corrector in the sampling process of diffusion models. I am wonder that if the negative results still holds in this setting?

---

> ### Author Rebuttal · Authors · 2026-03-31
>
> Dear reviewer, thank you for your careful reading and supportive review. We are very glad that you found our manuscript to “directly faces a important question” and that our “idea of exploiting high-dimensional concentration to "hide" score errors in low-probability regions of target distribution is clever and elegant”.
>
> We would like to respond to the two points you raised below. We believe these clarifications further strengthen the practical implications of our results. All sources here are the same as in our manuscript.
>
> **On score matching producing the behavior considered in the theory:** This is an excellent question and we thank the reviewer for raising it. We believe the behavior considered in our Theorem 3.7 can indeed occur with overfit score estimators, as supported by the synthetic simulations in Section 4.
>
> Here, we provide additional evidence that overfit score estimators can induce the “trapping” behavior around training points considered in Theorem 3.7. We measure the following properties of Langevin dynamics run with overfit score estimates produced as in Section 4:
>
> 1. Average displacement from initialization.
>
> 2. Average nearest neighbor distance to the training points.
>
> We compare the three algorithms from Section 4:
>
> -*Algorithm 1 (vanilla):* initializing Langevin at samples from $N(0, I_d)$.
>
> -*Algorithm 2 (fresh):* initializing Langevin at new samples from the target.
>
> -*Algorithm 3 (train):* initializing Langevin at the training samples.
>
> For both the Gaussian and GMM targets, Algorithm 3 exhibits *dramatically smaller displacement and nearest-neighbor distance* than Algorithm 1 and 2. This demonstrates that when Langevin is run with an overfit score estimate and initialized at training samples, it remains highly localized near the memorized points – exactly the effect of the construction in Theorem 3.7. As such, these simulations provide direct empirical evidence that the theoretical trapping mechanism can arise in practical settings.
>
> Results are averaged over 10 trials for $n = 13500$ generated samples; similar results hold for the other $n$ considered in Section 4.
>
>
> For Gaussian target:
>
> | | Algorithm 1 (vanilla) | Algorithm 2 (fresh) | Algorithm 3 (train) |
> |---|---|---|---|
> | Average displacement | $12.2987 \pm 0.1274$ | $12.3691 \pm 0.1573 $ | $ 5.8125 \pm 1.0386$ |
> | Average nearest neighbor distance | $8.9183 \pm 0.1212$ | $10.3104 \pm 0.1824$ | $5.0862 \pm 0.7707$ |
>
> For GMM target:
>
> | | Algorithm 1 (vanilla) | Algorithm 2 (fresh) | Algorithm 3 (train) |
> |---|---|---|---|
> | Average displacement | $11.3394 \pm 0.7604$ | $7.6374 \pm 0.1919 $ | $  0.3848 \pm 0.0040$ |
> | Average nearest neighbor distance | $9.3402 \pm 0.7390$ | $7.0880 \pm 0.2061$ | $0.3848 \pm 0.0040$ |
>
>
> **Negative results with underdamped Langevin:** This is also a very interesting question.
> If underdamped Langevin is used simply as the discretization of the Langevin SDE, our lower bounds still apply: they hold for *any* TV-close discretization (Remarks 3.3, 3.8). This includes underdamped Langevin, for example by Chewi et al. 2025 (see also Chewi 2025, Chapter 6.3).
> By contrast, if underdamped Langevin is used only as the corrector step inside a diffusion model, our negative results do not hold. In this case, the annealing path in the diffusion sampling process provides a robustness mechanism, and positive results under $L^2$ accuracy were established in e.g., Chen et al. 2023a.
>
> This distinction highlights the central message of our manuscript: the gap in robustness arises due to using Langevin dynamics rather than diffusion models as the high-level sampling strategy.
>
> We greatly appreciate your thoughtful and supportive feedback. We hope our response and additional simulations help clarify the practical significance of our results, and we would be very happy to discuss further. Thank you again for your time and consideration!

---

> > ### Author Rebuttal · Reviewer_MgRM · 2026-04-03
> >
> > I thank the authors for the thorough rebuttal.
> >
> > * The additional experiments on the trapping mechanism (Algorithm 3 vs. Algorithms 1and 2) directly validate the theoretical prediction of Theorem 3.7. It is very nice.
> >
> > * The clarification on underdamped Langevin, including the situation that is (a) sampler and (b) corrector address my concenrs.
> >
> > I will maintain my positive score 5.

---

### Decision · Program_Chairs · 2026-04-30

**Decision:**

Accept (regular)

**Comment:**

The paper shows that for score matching, even with L^2 score error that is exponentially small in dimension, Langevin dynamics initialized from training data (as opposed to held-out data) will not come close to the data distribution. The paper provided supporting experiments. Although the reviewers generally did not find the result surprising, as it confirmed intuition from earlier results, they found the formal theoretical result useful to establish.